# SWE-SQL: Illuminating LLM Pathways to Solve User SQL Issues in Real-World Applications

[α,ζ]**Jinyang Li*** [α,ζ]**Xiaolong Li*** [α,ζ]**Ge Qu*** [β]**Per Jacobsson** [ζ]**Bowen Qin** [ζ]**Binyuan Hui**
[ε,ζ]**Shuzheng Si** [α,ζ]**Nan Huo** [α,ζ]**Xiaohan Xu** [γ]**Yue Zhang** [α,ζ]**Ziwei Tang** [γ]**Yuanshuai Li**
[γ]**Florensia Widjaja** [γ]**Xintong Zhu** [γ]**Feige Zhou** [δ,ζ]**Yongfeng Huang**
[β]**Yannis Papakonstantinou** [β]**Fatma Ozcan** [γ,ζ]**Chenhao Ma**[†] [α,ζ]**Reynold Cheng**[†]
[α]HKU STAR Lab [β]Google Cloud [γ]CUHKSZ [δ]CUHK
[ε]THU [ζ]The BIRD Team
{jl0725,xiaolong,quge}@connect.hku.hk

🦉 https://bird-critic.github.io/

## Abstract

Resolution of complex SQL issues persists as a significant bottleneck in real-world database applications. Current Large Language Models (LLMs), while adept at text-to-SQL translation, have not been rigorously evaluated on the more challenging task of debugging on SQL issues. In order to address this gap, we introduce **BIRD-CRITIC**, a new SQL issue debugging benchmark comprising 530 carefully curated PostgreSQL tasks (BIRD-CRITIC-PG) and 570 multi-dialect tasks (BIRD-CRITIC-MULTI), which are distilled from authentic user issues and replayed within new environments to facilitate rigorous and contamination-free evaluation. Baseline evaluations on BIRD-CRITIC underscore the task's complexity, with the leading reasoning model O3-MINI achieving only 38.87% success rate on BIRD-CRITIC-PG and 33.33% on BIRD-CRITIC-MULTI. Meanwhile, realizing open-source models for database tasks is crucial which can empower local development while safeguarding data privacy. Therefore, we present **SIX-GYM** (**S**ql-f**IX**-Gym), a training environment for elevating the capabilities of open-source models specifically for SQL issue debugging. This environment leverages **SQL-Rewind** strategy, which automatically generates executable issue-solution datasets by reverse-engineering issues from verified SQLs. However, popular trajectory-based fine-tuning methods do not explore substantial supervisory signals. We further propose $f$-Plan Boosting, which extracts high-level debugging plans automatically from SQL solutions, enabling the teacher LLMs to harvest and produce 73.7% more successful trajectories for training. We integrate these components into an open-source agent, **BIRD-FIXER**. Based on Qwen-2.5-Coder-14B, BIRD-FIXER raises its success rate to 38.11% on BIRD-CRITIC-PG and 29.65% on BIRD-CRITIC-MULTI, surpassing many leading proprietary models such as Claude-3.7-Sonnet and GPT-4.1, marking a significant step toward democratizing sophisticated SQL-debugging capabilities for both research and industry.

## 1 Introduction

Relational Databases (RDBs) serve as the bedrock for data storage and information retrieval across countless modern applications, ranging from financial systems to web services and scientific research

---

*Equal contribution. † Corresponding author.

39th Conference on Neural Information Processing Systems (NeurIPS 2025).

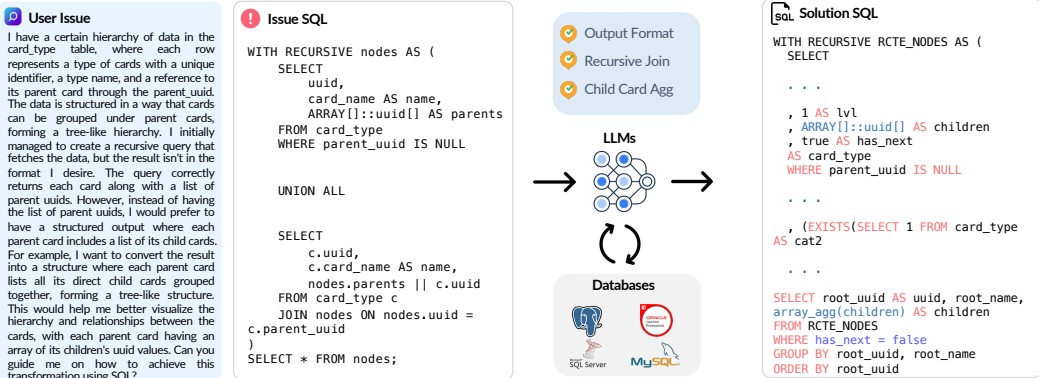

Figure 1: Illustration of the SQL issue debugging process in BIRD-CRITIC. It should start with a user issue query (left) and issue SQL query (center-left), LLMs will produce a corrected SQL solution (right) based on reasoning and interaction with the environment.

platforms [8, 34, 19, 35]. Structured Query Language (SQL), as the standard language for interacting with these systems, is thus a critical interface for data manipulation, querying, and administration [3, 2]. Despite its widespread adoption and apparent simplicity for basic operations, mastering SQL and troubleshooting complex queries or unexpected behaviors remains a significant challenge for users of all experience levels. The complexity of query semantics, diverse behaviors across different SQL operations (Create, Read, Update, Delete), evolving database features, and the need to understand underlying data schemas contribute to a steep learning curve and frequent user issues.

Resolving these SQL issues often demands considerable manual efforts, domain expertise, and time, representing a significant bottleneck in data-driven workflows and software development cycles [1, 25, 12, 40, 13]. Support forums, Q&A sites, and internal helpdesks, such as StackOverflow, are replete with user requests seeking assistance in debugging faulty queries, optimizing performance, or understanding why a query generates unexpected results. Therefore, automating this process holds huge value in improving productivity and reducing reliance on specialized human experts.

Recent advancements in Large Language Models (LLMs) have demonstrated remarkable capabilities in natural language understanding and code generation [6, 53, 7, 38, 50], notably achieving impressive results in converting natural language descriptions into SQL queries (text-to-SQL) [24, 45, 29, 22]. However, diagnosing and fixing existing incorrect or suboptimal SQL code presents more complex challenges. As shown in Figure 1, debugging such issues requires not only understanding the user's intent, often in a verbose and long-context description, but also analyzing the query logic underneath, identifying subtle errors, and intensively interacting with the database schema. Despite the practical importance of this task, the capabilities of current LLMs in SQL issue resolution have not been systematically investigated.

In this work, we are targeting to bridge this critical gap by two primary contributions. First, we present **BIRD-CRITIC**, a carefully curated benchmark built from authentic StackOverflow bug-fix threads. It comes in two subsets: (1) **BIRD-CRITIC-PG** with 530 PostgreSQL-only tasks, and (2) **BIRD-CRITIC-MULTI**, whose 570 tasks are distributed across 4 major dialects: PostgreSQL and MySQL as open-source databases, SQL Server and Oracle as community-friendly cloud-based platforms with free developer editions. Each task undergoes rigorous reconstruction where the underlying knowledge structures and debugging heuristics are extracted, and the scenario is reproduced within a controlled sandbox environment by new RDBs and conditions. This process ensures that tasks remain relevant while minimizing potential exposure to pre-training data. Furthermore, execution accuracy (EX) in standard text-to-SQL is inadequate for the diverse types of issues in BIRD-CRITIC, frequently leading to false negatives. Specifically, tasks involving database state changes, i.e., via Data Manipulation Language (DML) or Data Definition Language operations (DDL), frequently permit multiple functionally equivalent solutions that may differ syntactically or include non-impacting elements [52, 4]; reliance on strict EX matching would incorrectly penalize such valid SQL solutions. Therefore, each task is augmented with custom evaluation scripts containing specific test cases designed to evaluate functional correctness, enabling precise calculation of task success rates. Our baseline evaluations on BIRD-CRITIC underscore the complexity of SQL issue debugging, in which

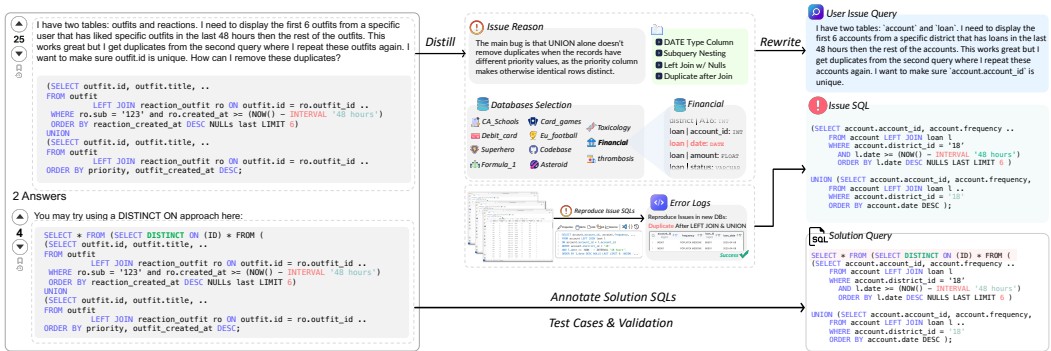

Figure 2: Example task structure within the BIRD-CRITIC benchmark, demonstrating the transformation from a user-reported issue and error SQL to a revised SQL solution.

even advanced reasoning models, O3-mini, only achieves a 38.87% success rate on BIRD-CRITIC-PG and 33.33% on BIRD-CRITIC-MULTI.

Second, inspired by prior work on code generation environments [28], we propose SIX-GYM (SQL-fIX-GYM), a training environment designed to enhance the SQL debugging capabilities of open-source models. A core innovation within SIX-GYM is the **SQL-Rewind** strategy, an automated methodology for generating large-scale, executable issue-solution datasets. This strategy operates by taking verified, correct SQL queries and systematically introducing plausible errors, effectively reverse-engineering realistic debugging scenarios. A common practice [28, 16] of such environments involves using an advanced teacher LLM to generate successful task execution trajectories for fine-tuning student smaller models. However, we find that this approach underutilizes the guidance available from ground-truth or reference solutions, potentially limiting the quantity and diversity of effective training trajectories. To address this, we introduce the **Functional Plan ($f$-plan) Boosting** strategy. This method first infers the underlying debugging logic by comparing the problematic SQL and the correct solution, representing this logic as a step-by-step pseudo-functional code plan. Afterwards, guided by this $f$-plan, a teacher LLM employs our designed agent scaffold, **SQL-ACT**, to execute the debugging task within the environment. This plan-guided approach generates a significant **73.7%** increase in more successful trajectories, providing richer data for fine-tuning open-source models, particularly smaller ones, to effectively interact with the database environment and debug complex SQL issues. The agent fine-tuned using this $f$-plan boosted data is termed BIRD-FIXER.

Our experiments demonstrate that BIRD-FIXER significantly enhances the performance of open-source models from various families. Notably, BIRD-FIXER fine-tuned on `Qwen-2.5-Coder-14B` achieves a 38.11% Success Rate (SR) on BIRD-CRITIC-PG and 29.65% on BIRD-CRITIC-MULTI, surpassing the performance of the highly capable models such as Claude-3.7-Sonnet and GPT-4.1. This result marks a significant advancement towards democratizing sophisticated SQL debugging capabilities for both research and practical industry applications.

## 2 Problem Definition

In this paper, we introduce a more complex but realistic task of SQL issue resolution. This task starts with a user-provided issue SQL query $\sigma_{\text{issue}}$, a natural language problem description $\mathcal{P}$ detailing the issue and intent, and the database schema $\mathcal{S}$. The goal is to generate a revised SQL query ($\sigma_{\text{pred}}$) that corrects the fault while preserving the user's intent. This mapping is:

$$\sigma_{\text{pred}} = f_\theta(\mathcal{P}, \mathcal{S}, \sigma_{\text{issue}}). \tag{1}$$

The desired output $\sigma_{\text{pred}}$ must satisfy the user's underlying intentions as inferred from the triplet $(\mathcal{P}, \mathcal{S}, \sigma_{\text{issue}})$. In BIRD-CRITIC, we annotated referenced ground-truth solution SQLs as $\sigma^*$ and develop tailored evaluation scripts (detailed in Section 3) for each task, enabling precise evaluation of the functional correctness of predicted solution SQLs.

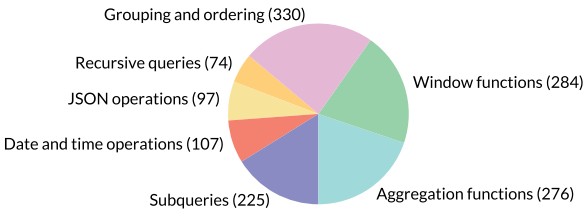

Figure 3: Distribution of issue categories in all BIRD-CRITIC, derived from an analysis of SQL usage in the real-world database applications. A detailed distribution is in Appendix E.

Table 1: Data Characteristics

| STATISTIC | PG | MULTI |
|---|---|---|
| **Total Issues** | 530 | 570 |
| # of query-like issues | 291 | 304 |
| # of management issues | 88 | 104 |
| # of personalization issues | 151 | 162 |
| user query length (mean/max) | 162.98/1046 | 165.75/1058 |
| issue SQL length (mean/max) | 133.29/1262 | 125.86/1254 |
| solution SQL length (mean/max) | 112.64/853 | 117.46/859 |
| # distinct test cases | 365 | 317 |
| # of preprocess SQLs | 643 | 571 |
| # of clean_up SQLs | 287 | 262 |
| inter-agreement | 94.53 | 92.98 |

## 3 BIRD-CRITIC Benchmark

**Annotator Group.** BIRD-CRITIC is developed via a multi-stage annotation converting raw user issues into executable, verifiable tasks. This involves two annotation groups: 1) 10 qualified database/SQL annotators, who pass strict entry test as detailed in Appendix B.1 and systematic training shown in Appendix B.2 to promise the quality of annotation; 2) 3 senior database experts/scientists for final data collection decision. This process is visually outlined in Figure 2.

**Environment Setup.** We leverage relational databases from the BIRD-SQL development set [24] chosen for its domain diversity across real data-science tasks (California Schools, Financial, Superheroes) and its permissive license. We migrate their original SQLite schemas to PostgreSQL, MySQL, SQL Server, and Oracle, four widely used production-grade dialects. During migration, we go beyond direct dialect translation by refining table and column names. We adjust data types and introduce guarded alterations to schema components to reduce potential information leakage (see Appendix A.2). To pair these databases with realistic debugging scenarios, we collect SQL issue queries from Stack Overflow, following a strict protocol shown in Appendix A.3.

**Issue Reproduction.** Following the initial collection of candidate issues, we start reproducing them in our environment in following produces as illustrated in Figure 2: (1) *Distilling Intent and Error:* Precisely identifying the user's underlying goal and the specific reason of the issue exhibited by $\sigma_{\text{issue}}$. The core reason of the issue is documented. (2) *Schema Mapping:* Assigning the issue to one of the adapted BIRD-SQL database schemas ($\mathcal{S}$) that provides a suitable context for the problem. (3) *Reproducibility Verification:* We adapt and execute $\sigma_{\text{issue}}$ against the chosen database, verifying through execution logs that the error appears as expected. This entire process transforms a potentially ambiguous web forum post into a standardized, reproducible problem instance $(\mathcal{P}, \mathcal{S}, \sigma_{\text{issue}})$ ready for solution annotation.

**Solution SQL & Evaluation Script Annotation.** Annotators carefully review the reproduced issue $(\mathcal{P}, \mathcal{S}, \sigma_{\text{issue}})$ and craft a new $\sigma^*$. This annotation requires ensuring that $\sigma^*$ can accurately fulfill the user's objective as inferred from $\mathcal{P}$ and the context of $\sigma_{\text{issue}}$. Also, to ensure robust evaluation, each task is annotated with evaluation scripts consisting of specific test cases written by Python and SQLs. Details can be found in Appendix C. We report the **Success Rate** (**SR %**), considering a task solved only when $\sigma_{\text{pred}}$ successfully passes **all** test cases in its evaluation script.

**Validation.** After annotation, BIRD-CRITIC undergoes cross-validation, with annotators exchanging data for review. This verification involves three steps: (1) enhancing test case functions with additional test cases for robust SQL code validation; (2) *red teaming* the SQL by introducing errors to make sure evaluation scripts can flag these errors. (3) Annotators first attempt to resolve disagreements through discussion. Persistent issues are escalated to the expert team for final determination, which may involve modification or rejection of the disputed annotation.

**Benchmark Statistics.** Table 1 summarizes the key properties of the BIRD-CRITIC benchmark, and Figure 3 visualizes the distribution of its underlying knowledge categories. The distribution of benchmark, is detailed in Appendix E. A side-by-side comparison with standard text-to-SQL benchmarks (Table 6 in Appendix E.1) exposes three distinctive challenges introduced by BIRD-CRITIC: non-query-like problems, multi-dialect complexities, and the most verbose but authentic user

queries. As far as we know, BIRD-CRITIC is the first debugging benchmark for SQL applications. These aspects establish BIRD-CRITIC as a crucial benchmark for rigorously evaluating LLM proficiency in solving authentic SQL issues.

# 4 SIX-GYM: An Automated SQL Debugging Environment for LLMs

This section introduces SIX-GYM, a dedicated training environment for enhancing the SQL debugging capabilities of LLMs. This environment is built upon **SQL-Rewind**, which is responsible for the automated generation of a comprehensive suite of SQL issue instances.

**Overview.** GYM-like datasets have proven effective for training LLMs as agents for complex tasks [28]. However, manually collecting and annotating these datasets is labor-intensive and difficult to scale, especially for debugging tasks. Thus, we introduce **SQL-Rewind**, which addresses this by inverting the debugging paradigm: starting with correct SQL queries ($\sigma^*$) and systematically introducing realistic issues to generate issue SQLs ($\sigma_{\text{issue}}$) and user issue query $\mathcal{P}$. This approach enables efficient creation of large-scale training data without human annotation. The pseudo-algorithm is shown in Appendix H.1.

**Solution SQL Collection.** We begin with raw StackOverflow issue data and enforce two principles against data overlap: (i) any issue used to construct BIRD-CRITIC tasks is excluded from SIX-GYM, and (ii) SQL-Rewind operates only on the 12 databases in the training databases of BIRD-SQL, while BIRD-CRITIC evaluation is confined to databases drawn solely from the BIRD-SQL dev set. We mine new candidate SQLs via rule-based regular expressions, then leverage `Gemini-2.0-Flash` to align table and column references to 12 databases in SIX-GYM, while preserving the original SQL's logical structure. To validate these adapted SQL queries as ground truth solutions, each was executed against its target database; only those queries that completed without error and yielded a non-null result were accepted into our final corpus of solution SQLs ($\sigma^*$).

**Synthetic Issue Generation and Automated Verification.** We employ `Gemini-2.0-Flash` to automate the entire process of issue reproduction and verification. Initially, the model summarizes issue reasons ($r_{issue}$) and modifies solution SQL ($\sigma^*$) to create issue SQL ($\sigma_{issue}$) guided by $r_{issue}$. Concurrently, it generates evaluation scripts $T$ comprising test cases designed to be passed by solution SQLs but failed by issue SQLs. The model then automatically validates whether the logic of triplet $\langle \sigma_{issue}, r_{issue}, \sigma^* \rangle$ is coherent and whether the evaluation script accurately identifies errors while allowing solution SQLs to pass. This validation process undergoes 3 iterative refinements; if the components are deemed compatible, the data is added to our collection.

**User Issue Query Generation.** Finally, we employ `Gemini-2.0-Flash` again to simulate a realistic user issue description $\mathcal{P}$. The generated $\mathcal{P}$ includes the user intent, issue description, and requirements. Each $\mathcal{P}$ must be logically consistent with $\langle \mathcal{S}, \sigma_{issue}, T, \sigma^* \rangle$. It undergoes up to 3 rounds of optimization by the model to reduce hallucinations. The resulting tuples are collected as final data. Using this SQL-Rewind strategy, we successfully generate approximately 3,301 high-quality synthetic data instances, forming a training environment we term **SIX-GYM**.

# 5 BIRD-FIXER: Elevating Open-Source LLMs to an SQL Issue Fixer

## 5.1 Agent Scaffold: SQL-ACT

ReAct [43] interleaves internal reasoning (thoughts $t_i$), external actions ($a_i$), and observations ($o_i$), and has proved highly effective for state-of-the-art code agents [28, 38, 39]. Building upon this paradigm, we introduce SQL-ACT, a specialized agent scaffold tailored for SQL tasks, particularly targeting challenges presented in benchmarks like BIRD-CRITIC. Unlike tool-based agents whose action space is restricted to a finite, hand-crafted set of operations, SQL-ACT treats arbitrary SQL commands as actions, dramatically enlarging the space of possible manipulations and enabling richer, more flexible debugging strategies.

At each step the agent emits a tuple $(t_i, \sigma_i, o_i)$, where $\sigma_i$ is the SQL statement executed at step $i$. The complete execution trajectory is therefore $\tau = ((t_1, \sigma_1, o_1), (t_2, \sigma_2, o_2), \ldots, (t_n, \sigma_n, o_n))$. As

Table 2: Success Rate (SR %) of different models on BIRD-CRITIC-PG and BIRD-CRITIC-MULTI, grouped by each issue and dialect categories. **Bold numbers indicate the highest score in each column**, and underlined numbers indicate the second highest. "Quer." = query-like issues, "Mana." = data-management issues, "Pers." = personalized-function issues. "PG." = PostgreSQL, "My." = MySQL, "Server" = SQL-Server.

| Model | BIRD-CRITIC-PG | | | | BIRD-CRITIC-MULTI | | | | |
|---|---|---|---|---|---|---|---|---|---|
| | Quer. | Mana. | Pers. | Overall | PG. | My. | Server | Oracle | Overall |
| *General-Purpose Models* | | | | | | | | | |
| Meta-Llama-3.1-8B | 18.21 | 22.73 | 11.26 | 16.98 | 13.04 | 13.27 | 21.43 | 3.06 | 12.81 |
| Phi-4 | 30.24 | 37.50 | 25.83 | 30.19 | 25.72 | 27.55 | 23.47 | 8.16 | 22.63 |
| Deepseek-V3 | 25.09 | 35.23 | 28.48 | 27.74 | 27.17 | 26.53 | 21.43 | 14.29 | 23.86 |
| Gemini-2.0-Flash | 27.84 | 44.32 | 29.14 | 30.94 | 27.54 | 22.45 | 31.63 | 7.14 | 23.86 |
| Meta-Llama-3.3-70B | 27.84 | 32.95 | 27.81 | 28.68 | 26.81 | 22.45 | 28.57 | 14.29 | 24.21 |
| Qwen2.5-Coder-32B | 31.62 | 38.64 | 24.50 | 30.75 | 28.26 | 24.49 | 30.61 | 9.18 | 24.74 |
| Claude-3.7-Sonnet | 27.15 | 43.18 | 35.10 | 32.08 | 32.61 | 30.61 | 21.43 | 18.37 | 27.89 |
| GPT-4.1 | 31.27 | 55.68 | 38.41 | 37.36 | 36.23 | 28.57 | 29.59 | 9.18 | 29.12 |
| *Reasoning Models* | | | | | | | | | |
| Gemini-2.0-Flash-Thinking | 27.15 | 53.41 | 33.11 | 33.21 | 28.99 | **35.71** | **37.76** | **19.39** | 30.00 |
| Claude-3.7-Sonnet-Thinking | 29.55 | 45.45 | 35.76 | 33.96 | 35.51 | 31.63 | 27.55 | 15.31 | 30.00 |
| O1-Preview-2024-09-12 | 29.90 | 53.41 | 37.09 | 35.85 | 40.94 | 33.67 | 33.67 | 11.22 | **33.33** |
| O3-Mini-2025-01-31 | **32.30** | **57.95** | **40.40** | **38.87** | **41.30** | 26.53 | 32.65 | 18.37 | **33.33** |

shown in Section 6.2, SQL-ACT is not only simpler to implement than TOOL-ACT but also delivers consistently higher accuracy in SQL issues solutions.

## 5.2 Trajectory Collection and Agent Fine-Tuning

$f$-**Plan Boosting.** The standard "gym-style" practice involves a strong teacher LLM on the environment and logs only those trajectories that reach the reference solution. In our experiments, running `Gemini-2.0-Flash` with SQL-ACT on SIX-GYM produces just 1,254 successful trajectories, which just utilizes 38.0% of the data.

To augment successful trajectories, we introduce $f$-**Plan Boosting**, a two-phase self-distillation loop:

**(1) Backward inference.** Given the problem $(\mathcal{P}, \mathcal{S}, \sigma_{\text{issue}})$ and its corrected query $\sigma^*$, the teacher annotates a step-by-step symbolic *functional plan* $F = (f_1, \ldots, f_k)$, where each $f_i$ represents an abstract debugging operation that maps $\sigma_{\text{issue}}$ toward $\sigma^*$. Since such plan contains few tokens yet exhibits more structured format, it is especially amenable to execution by LLMs [6, 18].

**(2) Forward validation.** Using only the context $(\mathcal{P}, \mathcal{S}, \sigma_{\text{issue}})$ and the candidate plan $F$, the teacher LLM regenerates a solution by SQL-ACT. The plan is accepted *iff* the regenerated solution SQL passes every test cases in $T$, producing a reliable pair $\langle (\mathcal{P}, \mathcal{S}, \sigma_{\text{issue}}), F \rangle$. After rollout we discard $F$ and retain only the executable trace $\tau' = \big((t_1, \sigma_1, o_1), \ldots, (t_n, \sigma_n, o_n)\big)$.

A single pass of $f$-Plan Boosting produces total 2,178 successful trajectories, an increase of **73.7%** over the vanilla collection pipeline, which we then use to fine-tune the open-source models via Low-Rank Adaptation (LoRA) [14].

**Generative Thought Mode (GTM).** The generalization of the agent can degrade when it predicts thoughts and actions jointly, because the model tends to overfit to the SQL patterns seen during fine-tuning. To counter this problem, we introduce a **Generative Thought Mode (GTM)**, which explicitly decouples the two predictions, akin to how Skip-gram in Word2Vec separates target and context words [26]. Let $M_O$ be the fine-tuned model, $M_B$ the original base model, and $H_{i-1} = ((t_1, \sigma_1, o_1), \ldots, (t_{i-1}, \sigma_{i-1}, o_{i-1}))$ the interaction history. During the inference step $i$, the fine-tuned model first proposes a thought–action pair $(t_i, \sigma_i) = M_O(H_{i-1})$, from which only the thought $t_i$ is extracted. The SQL action is then generated by the base model, $\sigma_i = M_B(H_{i-1}, t_i)$, leveraging its wide-coverage knowledge of diverse SQL dialects. GTM preserves the specialized debugging

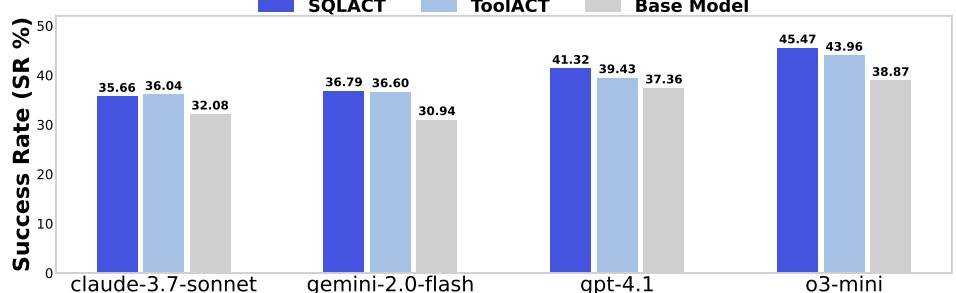

Figure 4: LLM agent performance for BIRD-CRITIC-PG. TOOACT employs constrained toolkit as actions, while SQLACT executes SQLs as actions.

logic learned by $M_O$, fully taking advantage of generative features of auto-regressive models [49], while mitigating overfitting of SQL patterns during training.

# 6 Experiments

## 6.1 SetUp

**Models.** We evaluate the performance of several popular and strong LLMs across two primary categories, including general-purpose models: `Gemini-2.0-Flash`, `GPT-4.1`, `Claude-3.7-Sonnet`, `Qwen-2.5-Coder-32B`, `Meta-Llama-3.1-8B`, `Meta-Llama-3.3-70B`, `Phi-4` and `DeepSeek-V3`. The second category consists of models specifically renowned for their advanced reasoning capabilities: `O3-mini`, `O1-preview`, `Gemini-2.0-Flash-Thinking`, and `Claude-3.7-Sonnet-Thinking`. The implementation details are in Appendix G.2.

**Advanced Agentic Methods.** Agentic workflows have shown considerable promise for addressing complex tasks. Accordingly, we also benchmark LLM agent performance on BIRD-CRITIC. Broadly, agentic systems can be classified into two main categories based on their action types. The first, which we term TOOL-ACT, involves agents employing pre-defined tools tailored to specific tasks. We implement Tool-Act guided by SOTA agents Spider-Agent [21] and InterCode [42] in SQL tasks. The second category, CODE-ACT [38], allows for more flexible, free-form actions where LLMs generate code to perform operations. In the context of this research, we implement a specific variant called SQL-ACT, where the LLMs generate SQL queries as their actions as introduced in Section 5.1.

## 6.2 Main Results

**Baseline Results.** An evaluation of mainstream Large Language Models (LLMs) on BIRD-CRITIC is detailed in Table 2. We can observe that:

(1) **Superior Performance of Reasoning-Oriented Models.** A clear performance advantage is evident for reasoning-oriented LLMs. These models surpass general-purpose counterparts by an average Success Rate (SR) of 6.13 % on PostgreSQL issues and 8.03 % on multi-dialect issues. This disparity underscores the computationally intensive, reasoning-driven nature of SQL-issue debugging, a task that demonstrably benefits from models capable of intermediate inferential steps.

(2) **Persistent Challenge Posed by SQL Issue Debugging.** Despite ongoing advancements in LLM capabilities, BIRD-CRITIC continue to present a considerable challenge. The top-performing model, `O3-Mini-2025-01-31`, achieves an overall SR of only 38.87% on PostgreSQL issues and 33.33% on multi-dialect issues, leaving large head-room for future research.

(3) **Heterogeneous Difficulty Across Issue Categories.**

An analysis of performance across distinct SQL issue categories reveals clear differences in difficulty. Issues related to data management, such as DML operations: insertions, deletions, updates, and DDL operations like schema modifications, are found to be relatively more manageable. On average, reasoning models achieved a 52.6% SR and general-purpose models a 38.8%

Table 3: Detailed comparison of BIRD-FIXER with other strong baselines on BIRD-CRITIC-PG and BIRD-CRITIC-MULTI. $\Delta$ shows relative improvement of BIRD-FIXER compared to base model.

| Model | BIRD-CRITIC-PG (SR %, ↑) | | | | BIRD-CRITIC-MULTI (SR %, ↑) | | | |
|---|---|---|---|---|---|---|---|---|
| | Base | SQL-ACT | BIRD-FIXER | $\Delta(\%)$ | Base | SQL-ACT | BIRD-FIXER | $\Delta(\%)$ |
| Llama-3.1-8B | 16.98 | 16.42 | **24.34** | +43.34 | 12.81 | 13.64 | **18.25** | +42.46 |
| Qwen-2.5-Coder-7B | 23.40 | 26.60 | **31.32** | +33.84 | 17.89 | 17.19 | **21.58** | +20.58 |
| Qwen-2.5-Coder-14B | 31.32 | 31.13 | **38.11** | +21.68 | 24.04 | 23.33 | **29.65** | +23.36 |
| Phi-4 | 30.19 | 29.43 | **38.11** | +26.23 | 22.63 | 19.80 | **27.89** | +20.58 |

SR in data management. Issues associated with Personalized functions also demonstrate moderate success rates. In contrast, Query-like issues present the greatest challenge for all LLMs.

These issues require an understanding of logical flaws within complex SELECT statements, particularly those involving joins, subqueries, aggregations, and conditional filtering. Unlike more standardized data management operations, SELECT queries exhibit remarkable diversity in their logic, structure, and intent, mostly reflected by the wide variety of underlying business requirements they serve, making their error patterns significantly harder to predict and correct. As evidenced in Figure 5, Query-like issues contain the most diverse functions, leading to the lowest performance of both general-purpose and reasoning models, which presents a strong negative correlation.

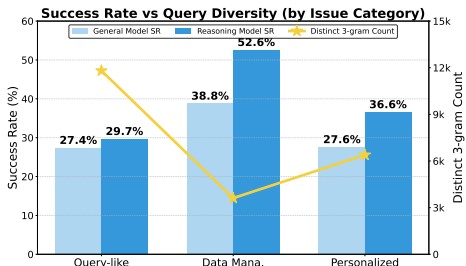

Figure 5: Success Rate vs Query Diversity (by Issue Category). It shows a strong negative correlation ($r$ = -0.89) between n-gram of tokens and model performance after normalization.

(4) **Dialect-Specific Performance Variations.** Model effectiveness exhibits notable dependency on the specific SQL dialect, as observed within the BIRD-CRITIC-MULTI. Specifically, `Gemini-2.0-Flash-Thinking` demonstrates the lower performance on PostgreSQL with a 28.99% SR. In contrast, it becomes the most proficient for SQL Server (37.76% SR), with a clear margin over other evaluated models in that dialect. Such variations are plausibly attributable to differential distributions of SQL dialects within the respective training corpora of these models, suggesting that the composition of training data significantly influences dialect-specific debugging capabilities.

(5) **Agentic Workflow Performance.** Figure 4 compares the performance of different LLM-based agents on BIRD-CRITIC-PG. The results show that agentic workflows markedly boost LLM accuracy on issue debugging tasks, which benefits from iterative interaction with its environment. Additionally, the SQL-ACT agent mostly outperforms the TOOL-ACT agent, suggesting that the richer, more flexible action space offered by SQL-ACT better equips LLMs to address the diverse and uncertain challenges encountered during debugging.

## 6.3 Performance Analysis of BIRD-FIXER

**Overall Performance of BIRD-FIXER.** Table 3 reports the performance gains achieved by BIRD-FIXER across three model families: Llama, Qwen, and Phi, which range from roughly 7B to 14B parameters. For each model, BIRD-FIXER delivers substantial improvements, demonstrating that the benefits of SQL-ACT + $f$-plan and SIX-GYM are architecture-agnostic and scalable. The table also exposes a limitation of small language models (SLMs) in agentic workflow only by inference: on several models, agent performance actually declines, suggesting that long, complicated interaction histories can overwhelm SLMs. By contrast, our methods equip these compact models with richer interaction capabilities, enabling them to navigate complex environments far more effectively. This benefit is especially valuable for privacy-sensitive SQL workloads: running a 7–14B parameter agent locally avoids any exposure of proprietary data to cloud services. Notably, BIRD-FIXER based on 14B base models, e.g., `Qwen-2.5-Coder-14B`, BIRD-FIXER presents competitive performance to

Table 4: Trajectory Generation Efficiency Comparison. **Baseline:** Standard SQL-ACT rollout with a single attempt (temperature=0). *f*-**Plan (Ours):** A single rollout guided by functional plans extracted from issue–solution pairs (temperature=0). **Rejection Sampling:** Up to 5 trials per instance (temperature=0.8), with early stopping when a successful trajectory is obtained. **Reject +*f*-Plan:** Combination of rejection sampling (up to 5 trials) with *f*-Plan guidance.

| Method | Max Tries | Successful Traj. | Avg Tries | DB Time (min) | Cost ($) |
|---|---|---|---|---|---|
| Baseline | 1 | 1,254 | 1.0 | 306 | 8.47 |
| *f*-Plan | 1 | 2,178 | 1.0 | 324 | 27.44 |
| Rejection Sampling | 5 | 1,910 | 4.2 | 1,377 | 108.05 |
| Reject + *f*-Plan | 5 | 2,560 | 1.7 | 810 | 41.16 |

`O3-mini` and outperforms the `Claude-3.7-Sonnet` agent on BIRD-CRITIC-PG, suggesting a promising path toward this goal of privacy while keeping effectiveness.

**Generalization to Multi-Dialect SQL Issue Debugging.** Although BIRD-FIXER is fine-tuned only on PostgreSQL trajectories within SIX-GYM, it generalizes robustly to other SQL dialects, as evidenced by the multi-dialect results in Table 3. That is because GTM elicits each model to produce a reusable debugging strategy trained in SIX-GYM while keeping pretrained knowledge of dialect variation. In conclusion, BIRD-FIXER exhibits strong cross-dialect reasoning without any extra data collection or further training, underscoring its practicality for heterogeneous database stacks.

## 6.4 Trajectory Sampling Comparison

To better illustrate the efficiency and effectiveness of *f*-Plan Boosting, we compare it against widely used trajectory augmentation approaches. We evaluate four strategies for trajectory generation, using `Gemini-2.0-Flash` as the teacher model on **SIX-GYM**.

As shown in Table 4, *f*-Plan Boosting yields $73.7\%$ more successful trajectories than the baseline while maintaining similar runtime and overhead. By contrast, rejection sampling increases success rates modestly but at the cost of $4.2\times$ more attempts and $4.5\times$ longer execution time. When combined, rejection sampling and *f*-Plan achieve the best overall trade-off, generating 2,560 trajectories with reduced average attempts (1.7) and a $62\%$ reduction in cost relative to rejection sampling alone. These results demonstrate that *f*-Plan provides an effective and efficient approach to trajectory augmentation during rollout in complex environments. Other detailed comparison can be found in Appendix E.3.

## 6.5 Ablation Study of BIRD-FIXER

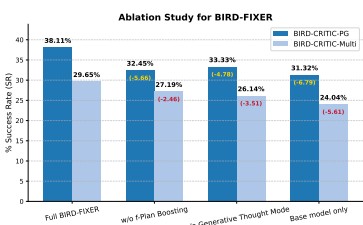

Figure 6: Ablation study of components in BIRD-FIXER.

Figure 6 shows the ablation study of BIRD-FIXER, highlighting:

**GTM (Generative Thought Mode):** Removing GTM causes the fine-tuned model $M_O$ to predict both thought and SQL action directly. The performance drop to 33.33% indicates that GTM effectively leverages the base model $M_B$ for SQL generation guided by $M_O$'s thought, mitigating overfitting to SQL patterns and better utilizing $M_B$'s broad SQL knowledge.

*f*-**Plan Boosting:** Using only trajectories from the vanilla collection pipeline reduces performance to 32.45% in BIRD-CRITIC-PG, highlighting *f*-Plan Boosting's importance in generating diverse, high-quality training trajectories crucial for complex reasoning tasks.

## 6.6 Error Analysis

To understand *how far* current LLM-based agents still are from fully resolving user-reported SQL issues, we sample 100 failed tasks from BIRD-CRITIC-PG by 4 agents based on: O3-mini, GPT-4.1, Claude-3.7-Sonnet, and BIRD-FIXER. It can be concluded that current agents exhibit four distinct error modes reflecting different levels of reasoning deficiency: **Projection Mismatch errors**

(**26.9%**), where models misinterpret output requirements by, for instance, adding unexpected columns or misapplying aggregations, suggesting limitations in semantic understanding of user intent and schema alignment; **Chain of Errors (27.3%)**, characterized by cascading failures due to partial problem resolution that overlooks dependent issues such as sequence updates accompanying primary key modifications, revealing difficulties in multi-step causal reasoning and consistency maintenance; The database engine only reports the most superficial issue, masking a deeper, dependent error that is the true root cause. For instance, a type mismatch error might be reported, but the underlying problem could be an incorrect join that brought together the wrong columns in the first place. **Incorrect Logic (44.5%)**, the most prevalent, highlighting fundamental misunderstandings of data structures or transformation methodologies, particularly in complex operations like `JSON` array manipulation, leading to syntactically plausible but semantically flawed SQL; and **Syntax Errors (29.3%)**, indicating technical implementation flaws such as type mismatches (e.g., `DATE` versus `TIMESTAMP`) or improperly formatted intervals, especially in specialized SQL contexts like recursive queries. The detailed examples for each category are in Figure 8. These findings highlight that future improvements should emphasize logical and schema-aware reasoning, cross-step dependency tracking, and dialect-robust SQL generation rather than mere syntactic refinement.

## 7   Related Work

**Large Language Models for Text-to-SQL.**   The automated conversion of natural language queries into Structured Query Language (SQL), known as Text-to-SQL, has garnered significant attention due to its practical utility in the era of big data [47, 41, 31, 15]. The advent of LLMs has notably advanced the capabilities in this domain. For instance, DIN-SQL [29], DAIL-SQL [9], TA-SQL [32], and Chase-SQL [30] have demonstrated SOTA performance on standard benchmarks like Spider [45] and BIRD [24], primarily by leveraging in-context learning with powerful foundation models like GPT-4. Also Supervised fine-tuning can fuel smaller LLMs towards stronger text-to-SQL parsers as evidenced by XiYanSQL[10], Arctic[48], OmniSQL [23], CodeS [22] , and SHARE [33]. Beyond direct generation, agentic workflows such as MAC-SQL [37], InterCode [42], which empowers LLMs to interact with database environments and gather contextual information, are pushing the boundaries of LLM cognition in handling complex and previously unseen databases. Concurrently, the field is evolving towards addressing more sophisticated, industry-relevant Text-to-SQL challenges. Initiatives like Beaver [5] and the Spider 2.0 [21] signify a shift from end-user focused queries to tasks requiring deeper BI knowledge and handling of larger schemas. This progression naturally leads to a critical, but underexplored, question: Can LLMs effectively diagnose and resolve issues within existing, user-provided SQL queries?

**LLMs for Program Repair.**   Program repair provides a complementary lens through which to evaluate and enhance the reasoning abilities of LLMs. At the function level, DEBUGBENCH [36] offers a multi-language suite that stresses fundamental programming logic. Repository-scale efforts such as SWE-BENCH [17] move closer to realistic software engineering, while follow-up studies, including SWE-LANCE [27] and MULTI-SWE [46] highlight the limitations of even sophisticated LLM-driven agents on complex, multi-language projects (e.g. Python, Java). Despite this rapid progress in general-purpose code fixing, *SQL-specific debugging remains largely unexplored*, even though databases are the backbone of most data-centric applications. To the best of our knowledge, our work is the first to formally cast SQL issue repair as a benchmark task, and to propose methods that adapt and augment open-source LLMs for automated SQL debugging.

## 8   Conclusion

We introduced **BIRD-CRITIC**, the first benchmark for SQL issue debugging tasks. Experiments show that SOTA LLMs solve fewer than 40% SR, underscoring the challenge. We also create **SIX-GYM**, an automated training environment which can produce thousands of high-quality agent trajectories without human annotation. Built on top of these trajectories, we proposed **SQL-Act**, a lightweight agent scaffold, and applied trajectory-level augmentation (*f-plan*) to fine-tune open-source LLMs, leading to the **Bird-Fixer**. Despite using only 7–14 B parameter backbones, BIRD-FIXER outperforms larger proprietary models and generalizes across four SQL dialects without additional training. Our research charts a path toward robust, real-world SQL issue debugging assistants.

# 9    Acknowledgments

We thank the anonymous reviewers and committees for their helpful comments, suggestions and organizations. We thank John Yang for early discussion and suggestions. Reynold Cheng, Jinyang Li, Xiaolong Li, Ge Qu, Nan Huo, Xiaohan Xu, and Ziwei Tang were supported by the Research Grant Council of Hong Kong (RGC Project HKU 17202325), the University of Hong Kong (Project 2409100399), and the HKU Faculty Exchange Award 2024 (Faculty of Engineering).

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

## Appendix Contents

# A    Environment Setup Details

## A.1    SQL Dialects Implementation

For the implementation of SQL dialects, we set up a sandbox environment using Docker[2] containers. This environment consists of four database containers and one evaluation container, all managed via a 'docker-compose.yml' configuration. The databases used in this setup include:

Table 5: SQL Dialects used in BIRD-CRITIC.

| Dialect | Version | URL |
|---|---|---|
| PostgreSQL | 14.12 | https://www.postgresql.org/ |
| MySQL | 8.4 Community Edition | https://www.mysql.com/ |
| Microsoft SQL Server | 2022 | https://www.microsoft.com/sql-server |
| Oracle | 19.3.0 Developer Edition | https://www.oracle.com/database/ |

Each of these databases is deployed in its own container, ensuring isolation and compatibility with the respective SQL dialects. The containers are connected through Docker Compose, allowing seamless interaction between the databases and the evaluation environment.

## A.2    Databases Migration & Modifications

Our initial setup begins with the BIRD-SQL development database, which is based on SQLite. The migration process is carried out using Navicat[3], a powerful database management tool. This tool is used to migrate the original SQLite databases to the four SQL dialects mentioned above.

After the migration, the schema structures of the databases are manually verified to ensure that they reflect the correct translations between different dialects. SQL queries, such as 'SELECT * FROM <table>', are executed to check data consistency and ensure that the migration retains the integrity of the original data. This step ensures that the translated databases can be used reliably for testing and evaluating SQL queries in the BIRD-CRITIC framework.

## A.3    Issue Collection Protocol

**User Issue Query Collection.**    StackOverflow, a prominent online Q&A platform for software development under a research-friendly license (CC BY-SA 4.0), is frequently utilized as a primary data source for code-related evaluation research, [20, 44, 11]. To ensure the issue quality, we pre-define a rigorous protocol based on 4 criteria: 1) presence of executable SQL code with identifiable errors or inefficiencies, 2) representation of significant database concepts from academic literature or real-world debugging practice, 3) appropriate complexity (queries exceeding 100 tokens or incorporating non-trivial function usage) and 4) sufficient contextual information to prevent ambiguity. We incorporate candidate issues that fulfilled at least 3 criteria, thereby assembling a representative collection of SQL challenges that authentically reflect the obstacles encountered in professional database application environments.

Annotators meticulously review the reproduced issue $(\mathcal{P}, \mathcal{S}, \sigma_{\text{issue}})$ and craft a new $\sigma^*$. This annotation requires ensuring that $\sigma^*$: (1) *Correctly Implements Intent:* Accurately fulfills the user's objective as inferred from $\mathcal{P}$ and the context of $\sigma_{\text{issue}}$. (2) *Resolves the Error:* Explicitly fixes the identified flaw(s) in $\sigma_{\text{issue}}$. (3) *Is Functionally Correct:* Executes successfully on the target database instance $D$ (conforming to $\mathcal{S}$) within the specified dialect and produces the expected, correct results. (4) *Adheres to Best Practices:* Solution SQLs should present a reasonably efficient and well-formed query. As shown in Figure 1, this results in a curated "Solution Query" ($\sigma^*$) paired with the user query and issue SQLs. Finally, to ensure robust evaluation, we annotate each task with evaluation scripts consisting of specific test cases written by Python and SQLs. Details can be found in Appendix C.

---

[2]https://www.docker.com
[3]https://www.navicat.com

**Docker Setup**

1. This tutorial guides you through setting up Docker on th
2. Note: if you are a Mac user, you can skip section 1 & 2
3. This tutorial is divided into 9 sections:
   a. WSL2
   b. Git Bash
   c. Docker
   d. Download from Google drive
   e. Python Env
   f. Stack Overflow Group
   g. GitHub Issue Group
   h. Git Bash in VScode
4. **Use VSCode for python code**
5. Last modified date: 2024-12-06

**Data Annotation Instruction**

1. In this tutorial, you will learn how to reproduce the Stack Over CRITIC Enviornment.
2. This tutorial is divided into 8 sections:
   a. Python Utils
   b. CREATE Case
   c. SELECT Case
   d. UPDATE Case
   e. DROP Case
   f. INSERT Case
   g. ALTER Case
   h. Recursive function usage, JSON, optimization
3. YOU MUST SETUP THE Docker ENV BEFORE WORKING ON T
   a. Check this slide to get an overall idea about this project an
4. **Use VSCode for python code, IDE/Python for SQL query**
5. Last modified date: 2025-01-21

**SQL Debugging Entry Exam for BIRD-CRITIC Annotators**

**Exam Purpose and Structure**

This entry exam evaluates your proficiency in SQL issue identification, debugging techniques, and solution implementation. You must successfully complete all 10 challenges to qualify as a BIRD-CRITIC annotator. You have one week to complete this exam.

**Environment Setup Requirements**

YOU MUST SETUP THE Docker ENV BEFORE WORKING ON THIS EXAM
- Use the provided Docker environment for consistent evaluation
- Setup instructions can be found in the Docker Setup tutorial
- Use VSCode for Python code, IDE/Python for SQL queries

**Exam Structure**

This exam consists of 10 SQL debugging challenges across different categories:
1. **Basic SELECT Query Debugging** (PostgreSQL)
   ◦ Task: [Task description will be provided]
   ◦ Schema: [Schema information will be provided]
   ◦ Issue SQL: [Problem SQL will be provided]
   ◦ Your task: Identify issues, fix the query, and explain your solution

Figure 7: Examples of training materials by screenshots for BIRD-CRITIC annotators. Left: Docker setup instructions for creating the standardized annotation environment. Middle: Data annotation tutorials with detailed procedures for reproducing SQL issues. Right: Entry examination outline used to evaluate annotator proficiency across various SQL debugging challenges.

# B  Annotator Qualification

## B.1  Annotator Entrance Test

To ensure high-quality annotations for the BIRD-CRITIC benchmark, we implemented a rigorous training process for all annotators. Each potential annotator underwent a comprehensive training program before contributing to the benchmark creation.

## B.2  Training Tutorial

Annotators participated in an intensive tutorial program covering essential aspects of SQL issue debugging, including:

- Database environment setup
- Database schema analysis and comprehension
- SQL error identification patterns and common debugging approaches
- Systematic issue reproduction techniques
- Solution validation and evaluation script development
- Best practices for creating test cases across different SQL dialects (PostgreSQL, MySQL, Oracle, and SQL Server)

The training materials included detailed documentation, practical examples, and hands-on exercises that mirrored the complexity and diversity of real-world SQL issues. Annotators were introduced to the specific annotation workflow required for BIRD-CRITIC benchmark creation.

## B.3  Qualification Test

Following the week-long training phase, each candidate annotator was required to complete a qualification test consisting of ten representative SQL issue debugging tasks.

For each task, candidates had to:

1. Correctly identify the underlying issue in the problematic SQL
2. Reproduce the issue in the controlled environment
3. Develop a solution SQL that resolved the identified problems
4. Create comprehensive test cases to validate solution correctness
5. Document their reasoning and approach

Only candidates who successfully completed all ten tasks with satisfactory quality were approved as annotators for the BIRD-CRITIC benchmark. This stringent qualification process ensured that all annotators met the high standards required for creating a robust and trustworthy benchmark.

The qualification test success rate was approximately 90%, indicating the effectiveness of our tutorial materials and instruction program in preparing candidates for SQL issue debugging tasks. All annotators who contributed to the final BIRD-CRITIC benchmark successfully passed this qualification process.

## C   Evaluation Script Details

To rigorously evaluate the correctness and suitability of generated SQL solutions ($\sigma_{\text{pred}}$), particularly in the context of issue resolution, evaluation methodologies must extend beyond superficial syntactic checks or simple result set comparisons. We annotate each task with specific test case functions, which encompass four categories of SQL issue types in BIRD-CRITIC:

- **Query-like Issues:** Predominantly for conventional `SELECT` queries. Given that BIRD-CRITIC already provides issue SQLs that deliver original user intents, the solution SQLs must preserve these intentions while addressing identified problems. This protocol assesses correctness by executing $\sigma_{\text{pred}}$ and the ground-truth $\sigma^*$ on the database instance $D$ and verifying the semantic equivalence of their result sets, typically accommodating variations in tuple ordering unless explicitly constrained by the task specifications.

- **Management Issues:** Essential for tasks involving Data Manipulation Language (DML: `UPDATE`, `INSERT`, `DELETE`), Data Definition Language (DDL: `CREATE`, `ALTER`), Data Control Language (DCL: `GRANT`, `REVOKE`), or complex multi-step procedures. For these cases, domain experts manually design test cases to ensure that the results executed by $\sigma_{\text{pred}}$ fulfill the specified user requirements.

- **Personalization Issues:** For tasks imposing specific syntactic or semantic constraints on the solution (e.g., mandatory use of certain SQL features, avoidance of others, derived from the problem description $\mathcal{P}$), this category extends the test case functions of the previous two categories while enforcing additional compliance criteria.

## D   Evaluation Metrics

In BIRD-CRITIC, we adopt the **Task Resolution Success Rate (SR %)** as metric. This metric measures the percentage of tasks for which a model generates a SQL solution $\sigma_{\text{pred}}$ that successfully passes the **all** curated test cases in the evaluation script. Formally, let $N$ be the total number of tasks in the evaluation set, and let $T_i$ represent the dedicated evaluation script designed for task $i$. A generated solution $\sigma_{\text{pred},i}$ for task $i$ is considered successful if and only if $T_i(\sigma_{\text{pred},i})$ returns a passing outcome (returns `True`). The overall Success Rate is then calculated as:

$$\text{SR} = \frac{1}{N} \sum_{i=1}^{N} \mathbb{I}(T_i(\sigma_{\text{pred},i}) = \texttt{True})$$

where $\mathbb{I}(\cdot)$ denotes the indicator function, evaluating to 1 if the condition is true and 0 otherwise. This metric directly leverages the outcomes of our comprehensive, category-aware test case framework. Since each test function $T_i$ is tailored to the specific nature of the user's issue, evaluating semantic equivalence of results (Soft EX), correctness of database state transitions, adherence to explicit constraints via parsing as appropriate, the SR provides a holistic measure of a model's capability. It assesses the model's ability to generate solutions that are not merely executable, but are functionally correct and contextually appropriate for resolving the specific problem presented in the task instance $(\mathcal{P}, \mathcal{S}, \sigma_{\text{issue}})$. We argue that this success rate provides a more rigorous and practically relevant assessment of SQL issue resolution capabilities compared to metrics focused solely on execution or partial component matching.

Table 6: Data statistics of features in BIRD-CRITIC compared to related benchmarks. [†]: Results taken from public available Spider 2.0 Lite Gold SQL. EM refers to the Exact Match, EX refers to Execution Accuracy, and PCM-F1 refers Partial Component Match F1.

| Dataset | # Eval | # Toks. / Q | # Toks. / SQL | Evaluation Metric | Non Query-like | Multi-Dialect |
|---|---|---|---|---|---|---|
| Spider 1.0 | 1,034 | 14.28 | 30.18 | EM/EX | ✗ | ✗ |
| SEDE | 857 | 14.34 | 101.3 | PCM-F1 | ✗ | ✗ |
| BIRD-SQL | 1,543 | 18.36 | 50.01 | EX | ✗ | ✔ |
| Spider 2.0† | 547 | 61.93 | 412.37 | EX | ✗ | ✔ |
| BEAVER | 203 | 59.27 | 538.13 | EX | ✗ | ✗ |
| BIRD-CRITIC PG | 530 | 307.35 | 111.47 | Test Cases | ✔ | ✗ |
| BIRD-CRITIC MULTI | 570 | 296.27 | 112.64 | Test Cases | ✔ | ✔ |

# E  More Statistics

## E.1  Comparison of BIRD-CRITIC with other conversational Text-to-SQL benchmarks

This section compares BIRD-CRITIC with other benchmarks, highlighting its advantages in handling significantly longer user queries and supporting non-query-like SQL statements (e.g., DML, DDL), which present additional challenges. Additionally, the custom-designed test cases ensure a faithful evaluation of SQL solutions, while the multi-dialect support enables more comprehensive evaluation across diverse environments

## E.2  Detailed Statistics of BIRD-CRITIC-MULTI

This section focuses on the detailed statistics of the BIRD-CRITIC-MULTI dataset, emphasizing its support for multiple SQL dialects and showcasing the distribution of query types, SQL issues, and test cases across diverse dialects.

Table 7: Statistics grouped by Category and Dialect

| | Count | Query | | Issue SQL | | Solution SQL | | Test Cases | |
|---|---|---|---|---|---|---|---|---|---|
| **Category** | | Mean | Max | Mean | Max | Mean | Max | Mean | Max |
| Query | 304 | 179.12 | 1058 | 168.20 | 1262 | 126.72 | 853 | 80.29 | 134 |
| Management | 104 | 141.44 | 519 | 68.80 | 267 | 102.76 | 578 | 189.53 | 733 |
| Personalization | 162 | 146.43 | 528 | 100.21 | 1073 | 113.01 | 778 | 160.50 | 517 |
| **Dialect** | | | | | | | | | |
| PostgreSQL | 276 | 152.61 | 1058 | 78.17 | 1073 | 103.45 | 578 | 151.30 | 733 |
| MySQL | 98 | 152.86 | 435 | 65.12 | 230 | 93.40 | 778 | 93.34 | 281 |
| Oracle | 98 | 171.52 | 421 | 265.36 | 1262 | 155.92 | 853 | 93.95 | 342 |
| SQLServer | 98 | 192.17 | 403 | 214.31 | 798 | 145.89 | 542 | 95.57 | 459 |

## E.3  Quality Validation of SIX-GYM

This section validates the quality of synthetic data generated by **SQL-Rewind** by comparing **SIX-GYM** with the manually curated BIRD-CRITIC-PG benchmark. Table 8 demonstrates that our synthetic dataset exhibits comparable complexity and diversity across multiple dimensions, including similar distribution of complex operations, higher SQL diversity ratio, and comparable query lengths to human-annotated challenging data.

Table 9 further breaks down performance by SQL complexity. The benefits of $f$-Plan scale with difficulty: while gains over rejection sampling are modest on simple queries (+7.5 points), they grow dramatically on complex queries with $5+$ clauses (+29.2 points). $f$-Plan also resolves instances unsolved by either baseline or rejection sampling, particularly those with high keyword diversity and nested operations. These results highlight that $f$-Plan narrows the search space through structured

Table 8: Data Statistics Comparison between SIX-GYM and BIRD-CRITIC-PG [†]: Diversity Ratio = Unique 3-grams / Total 3-grams.

| Dimension | BIRD-CRITIC-PG | SIX-GYM |
|---|---|---|
| User Query Length (mean/max) | 162.98/1046 | 171.1/882 |
| Issue SQL Length (mean/max) | 133.29/1262 | 110.2/1089 |
| Solution SQL Length (mean/max) | 112.64/853 | 94.8/772 |
| SQL Keywords Coverage | 165 | 157 |
| Complex Operations (%) | 54.5 | 54.3 |
| Multi-clause Queries (%) | 59.4 | 61.2 |
| SQL Diversity Ratio† | 0.728 | 0.750 |

debugging plans, providing explicit guidance that is especially valuable when random exploration becomes ineffective.

Table 9: Success Rate by SQL Complexity

| Issue SQL Complexity | Baseline | Reject | $f$-Plan |
|---|---|---|---|
| Simple (1-2 clauses) | 52.3% | 62.8% | **70.3%** |
| Medium (3-4 clauses) | 38.7% | 58.8% | **69.4%** |
| Complex (5+ clauses) | 19.4% | 37.1% | **66.3%** |
| High Keyword Diversity (10+) | 24.1% | 35.6% | **54.3%** |
| Nested Operations (2+ levels) | 21.8% | 36.8% | **49.2%** |

# F  Error Analysis Details

Figure 8 shows examples for each error type, along with an analysis of why the LLM-generated SQL failed the issue SQL query resolution.

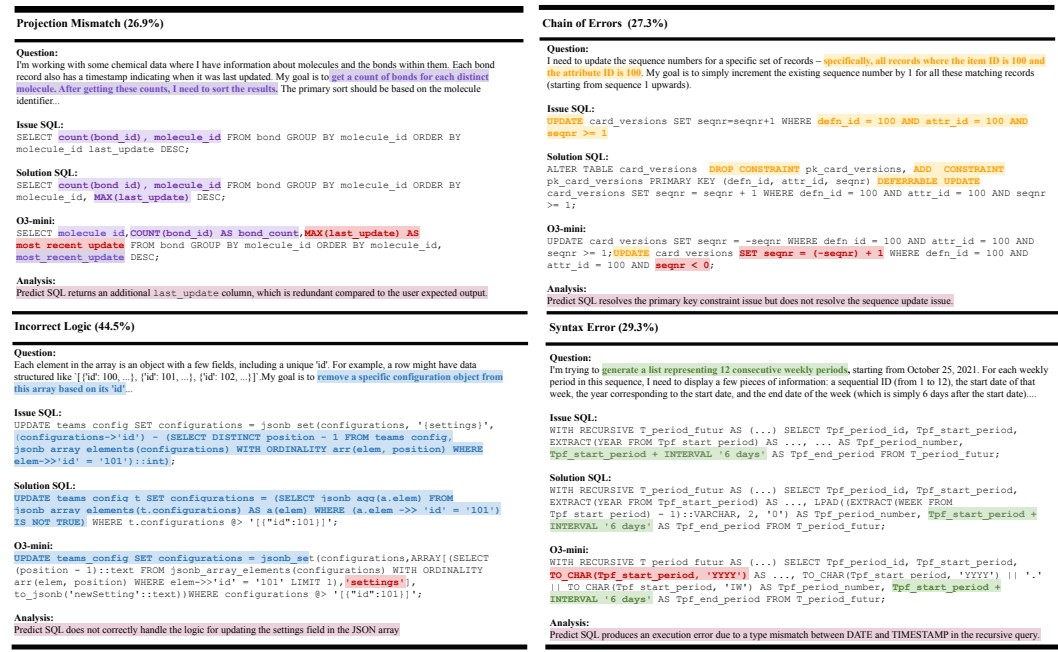

Figure 8: Detailed Error Analysis

# G Experiment Details

## G.1 Alias of LLMs

The following aliases are used for the models in this work:

- Claude-3.7-Sonnet: `claude-3-7-sonnet-20250219`
- Claude-3.7-Sonnet-Thinking: refers to `claude-3-7-sonnet-20250219` with extended thinking
- O3-Mini: `O3-Mini-2025-01-31`
- O1-Preview: `O1-Preview-2024-09-12`
- GPT-4.1: `gpt-4.1-2025-04-14`
- Gemini-2.0-Flash: `gemini-2.0-flash`
- Gemini-2.0-Flash-Thinking: `gemini-2.0-flash-thinking-exp-01-21`
- deepseek-v3: `deepseek-chat`
- deepseek-r1: `deepseek-reasoner`

All open-source models are downloaded from Hugging Face[4]:

- Llama: `Meta-Llama-3.1-8B-Instruct`, `Meta-Llama-3.3-70B-Instruct`
- Qwen-Coder: `Qwen2.5-Coder-7B-Instruct`, `Qwen2.5-Coder-14B-Instruct`, `Qwen2.5-Coder-32B-Instruct`
- Phi: `Phi-4`

## G.2 Model Implementation Details

For inference with proprietary models, we use official API providers, including OpenAI (`https://openai.com/`), Anthropic (`https://www.anthropic.com/`), Google (`https://gemini.google.com/`), and Deepseek (`https://www.deepseek.com/`). The total API cost for proprietary models is around $200 USD.

For open-source models, we fine-tune all our models using the LlaMa-Factory library [51] (version 0.9.2) `https://github.com/hiyouga/LLaMA-Factory` with LoRA [14]. All our experiments are conducted on $8\times$H100 GPU with 80GB memory. We set the low-rank dimensions as 8, the learning rate as $5e^{-5}$, and the batch size as 4. The specific training hours for each backbone model are shown in Table 10. We use VLLM[5] (version 0.6.4.post1) to perform inference. We set the temperature as 0.1, the top p as 0.95, and the maximum input token length as 8000. We report the experimental results as the average of five repeated trials. The total GPU hours spent on inference are approximately 20 hours.

Table 10: GPU hours spent to train each backbone model.

| Model | GPU Hours |
|---|---|
| Meta-Llama-3.1-8B | 24.88 |
| Qwen2.5-Coder-7B | 22.00 |
| Qwen2.5-Coder-14B | 35.93 |
| Phi-4 | 31.42 |

## G.3 Agent Implementation Details

All agent designs follow the ReAct framework [43], which uses interleaving Thought, Action, Observation steps. Specifically:

---

[4]`https://huggingface.co/`
[5]`https://docs.vllm.ai/en/latest`

- **SQL-ACT**: The action is the freedom to execute any executable SQL query.
- **Tool-ACT**: Actions are predefined and include:
  - Schema Inspection: Reveals table/column information.
  - Sample Data: Previews example rows from a table.
  - Solution Query: The final, correct SQL query that resolves the user's issue.

# H  Algorithm

## H.1  SQL Rewind Algorithm

We formalize the end-to-end SQL-Rewind pipeline in Algorithm 1, outlining each stage from raw post extraction to the construction of high-quality training tuples.

---

**Algorithm 1** Automatic construction of SIX-GYM training instances with **SQL-Rewind**.

---

**Require:** $\mathcal{D}_{\text{raw}}$ (Stack Overflow posts), $\mathcal{W}$ (training databases); $target\_size$; $max\_iter$
**Ensure:** $|\mathcal{G}| \geq target\_size$
  **procedure** SQL_REWIND
      $\mathcal{G} \leftarrow \emptyset$                                                           $\triangleright$ collected training tuples
      **for** each $post$ in $\mathcal{D}_{\text{raw}}$ **do**
         **if** OVERLAP_WITH_BIRD_CRITIC($post$) **then**
            **continue**
         **end if**
         $C \leftarrow$ EXTRACT_SQL($post$)                            $\triangleright$ regex extraction
         **for** each $sql$ in $C$ **do**
            **for** each $db$ in $\mathcal{W}$ **do**
               $sol\_sql \leftarrow$ ADAPT_SCHEMA($sql, db$)
               **if** EXEC_OK($sol\_sql, db$) **then**           $\triangleright$ issue synthesis and verification
                   **for** $i \leftarrow 1$ **to** $max\_iter$ **do**
                      $(\sigma_{\text{issue}}, r_{\text{issue}}, T) \leftarrow$ GEN_ISSUE($sol\_sql, db$)
                      **if** VALIDATE($\sigma_{\text{issue}}, r_{\text{issue}}, T, sol\_sql, db$) **then**
                          **break**
                      **end if**
                 **end for**
                 **if** validation failed **then continue**
                 **end if**                            $\triangleright$ user query generation
                 **for** $j \leftarrow 1$ **to** $max\_iter$ **do**
                      $\mathcal{P} \leftarrow$ GEN_USER_QUERY($\sigma_{\text{issue}}, r_{\text{issue}}, T, db$)
                      **if** CONSISTENT($\mathcal{P}, \sigma_{\text{issue}}, T, sol\_sql$) **then**
                          **break**
                      **end if**
                 **end for**
                 **if** consistency failed **then continue**
                 **end if**
                 $\mathcal{G} \leftarrow \mathcal{G} \cup \{\langle db.\mathcal{S}, \mathcal{P}, \sigma_{\text{issue}}, T, sol\_sql \rangle\}$
                 **if** $|\mathcal{G}| \geq target\_size$ **then**
                     **break all loops**
                 **end if**
               **end if**
            **end for**
         **end for**
      **end for**
      **return** $\mathcal{G}$
  **end procedure**

---

## H.2  BIRD-FIXER Algorithm

**Algorithm 2** BIRD-FIXER: Functional planning, backward inference, and forward validation for SQL issue fixing.

---

**Require:** $\mathcal{P}, \mathcal{S}, \sigma_{\text{issue}}; \sigma^*, T; F = (f_1, \ldots, f_k)$
**Ensure:** Trajectory $\tau' = ((t_1, \sigma_1, o_1), \ldots, (t_n, \sigma_n, o_n))$
  **Function:** BIRD-FIXER
  **procedure** FUNCTIONALPLAN
      Annotate symbolic functional plan $F = (f_1, \ldots, f_k)$ from teacher LLM
      **for** each $f_i$ in $F$ **do**
         $f_i$ represents an abstract debugging operation mapping $\sigma_{\text{issue}}$ to $\sigma^*$
      **end for**
  **end procedure**
  **procedure** BACKWARDINFERENCE
      Given the problem $(\mathcal{P}, \mathcal{S}, \sigma_{\text{issue}})$ and the corrected query $\sigma^*$
      Generate a step-by-step functional plan $F = (f_1, \ldots, f_k)$
      $F$ is annotated by the teacher LLM to map $\sigma_{\text{issue}}$ to $\sigma^*$
  **end procedure**
  **procedure** FORWARDVALIDATION
      Using $(\mathcal{P}, \mathcal{S}, \sigma_{\text{issue}})$ and candidate plan $F$
      Regenerate solution using SQL-ACT with teacher LLM
      **if** Regenerated SQL passes all test cases in $T$ **then**
         Accept $F$
         Retain executable trace $\tau' = ((t_1, \sigma_1, o_1), \ldots, (t_n, \sigma_n, o_n))$
      **else**
         Discard plan $F$
      **end if**
  **end procedure**

---

# I  Limitation And Future Work

Our work primarily focuses on SQL content and knowledge by simplifying the impact of external workflows through containerized Docker environments. Workflow operations such as file reading and editing represent important considerations for future development in BIRD-CRITIC 1.5. Actually, We conducted preliminary experiments on models performing workflow-integrated content-based tasks, where LLMs not only check and revise SQL issues but also save results to files. This integration resulted in substantial performance drop, with success rates dropping from approximately 30% to 10%. However, we prioritize SQL knowledge improvement in this work since significant opportunities for advancement remain in this domain.

Similar to most complex task evaluations [53], BIRD-CRITIC employs single-turn evaluation while striving to make task descriptions as clear as possible. However, real-world applications typically require crucial interaction between users and agents since most users cannot articulate their intents or queries with complete clarity and may need multi-turn interactions for clarification or additional information processing. Our recent work, BIRD-Interact[6], evaluates text-to-SQL performance of LLM agents through dynamic interaction by multi-turn conversational and agentic interactions. Future work will extend BIRD-CRITIC to incorporate dynamic user-SQL debugging processes, better simulating the complexity of real-world agent-human interactions.

# J  Broader Impact

Our work presents an approach to training open-source models specifically designed for debugging SQL issues. Additionally, we introduce a workflow for constructing robust benchmarks from diverse open platforms, such as StackOverflow, through a reproducible loop to mitigate potential data leakage. Furthermore, our research primarily targets technical SQL knowledge within the programming domain. Thus, it does not directly engage with or pose risks concerning broader societal issues.

---

[6]https://bird-interact.github.io/

# K  Prompt

**Baseline Prompt for resolving SQL issues with an LLM**

You are a SQL assistant. Your task is to understand user issue and correct their problematic SQL given the database schema. Please wrap your corrected SQL with ```sql\n[Your Fixed SQL]\n``` tags in your response.

**Database Schema**:
{SCHEMA}

**User issue**:
{USER_ISSUE}

**Problematic SQL**:
{ISSUE_SQL}

**Corrected SQL**:

## Prompt used to generate Thought

Interact with the `"{db_id}"` database using PostgreSQL to solve the user issue. You will be given the following information:
1. **Database schema**: complete `CREATE TABLE ...` DDL.
2. **User Issue**: a natural language description of the desired outcome or the current bug.
3. **Problematic SQL**: the query (or queries) that presently fail to meet the requirement.

Use interleaving Thought, Action, Observation steps.
**Thought** can reason about the possible errors or other information you think you need for debugging about the current situation. For instance, it could be:

- Diagnosis of the bug you see in the current query.

- Hypotheses you want to confirm (e.g., Maybe the join is missing a date filter).

- Reasoning that led you to the next SQL step (checking row counts, inspecting NULLs, etc.).

- A brief plan for what you will try next.

**Action** can only be the executable PostgreSQL SQL. The **Observation** would be the execution results feedback from the environment.
Wrap your thought in the `<thought>[Your Thought]</thought>` tag and your action in `<action>[Executable SQL]</action>`.
The input for you is as follows:
**Database Schema**
{SCHEMA}

**User Issue**
{USER_ISSUE}

**Problematic SQL**
{ISSUE_SQL}

**Important Rules:**

- **MOST IMPORTANT:** Wrap your thought in the `<thought>[Your Thought]</thought>` tag and your action in the `<action>[Executable SQL]</action>` tag.

- The action inside the `<action></action>` tags must be pure PostgreSQL statements that can be executed directly, without any comments or needs for additional post-processing.

Now generate the thought and action of the next round given the trajectory history and the input. You still have {turn} turns left.
**React**
{history}

`<thought>`

## Prompt used to generate Action

Interact with the `"{db_id}"` database using PostgreSQL to solve the user issue. You will be given the following information:
1. **Database schema**: complete `CREATE TABLE ...` DDL.
2. **User Issue**: a natural language description of the desired outcome or the current bug.
3. **Problematic SQL**: the query (or queries) that presently fails to meet the requirement.

Use interleaving Thought, Action, Observation steps.
**Thought** can reason about the possible errors or other information you need for debugging about the current situation. For instance, it could be:

- Diagnosis of the bug you see in the current query.

- Hypotheses you want to confirm (e.g., Maybe the join is missing a date filter).

- Reasoning that led you to the next SQL step (checking row counts, inspecting NULLs, etc.).

- A brief plan for what you will try next.

**Action** can only be the executable PostgreSQL SQL according to the corresponding thought. The **Observation** would be the execution results feedback from the environment.

Your task is to generate the action for the current round thought given the react history. Wrap your action in `<action>[Executable SQL]</action>`. If you think the debugging process is done, just output `<action>[DONE]</action>` as the action.

The input for you is as follows:
**Database Schema**
{SCHEMA}

**User Issue**
{USER_ISSUE}

**Problematic SQL**
{ISSUE_SQL}

**Important Rules:**

- **MOST IMPORTANT:** Wrap your action in `<action>[Executable SQL]</action>`.

- The action inside the `<action></action>` tags must be pure PostgreSQL statements that can be executed directly, without any comments or needs for additional post-processing.

- If you believe the debugging process is finished, output `<action>[DONE]</action>` as the action for this turn.

Now generate the action of this round given the trajectory history and current thought. Generating multiple rounds at once is NOT ALLOWED! You still have {turn} turns left.
**React**
{history}

`<action>`

## Prompt used to generate Corrected SQL

You are a text-to-SQL expert. You will be given the following information:
1. **Database schema**: complete `CREATE TABLE ... ` DDL.
2. **User Issue**: a natural language description of the desired outcome or the current bug.
3. **Problematic SQL**: the query (or queries) that presently fails to meet the requirement.
4. **React Thought Chain**: A history of your prior debugging iterations, formatted as a sequence of thought → action → observation tuples. Each tuple is separated from the next by a blank line (
`n`
`n`).

- **Thought** - Your reasoning: hypotheses about errors, assumptions, or additional data requirements.

- **Action** - A pure PostgreSQL statement executed to test or correct the issue.

- **Observation** - The execution result returned by the database engine.

Your task is to understand all these contents and generate the final PostgreSQL that could accurately solve the user issue.
The input is as follows:
**Database Schema**
{SCHEMA}

**User Issue**
{USER_ISSUE}

**Problematic SQL**
`sql_list = {ISSUE_SQL}`

**React Thought Chain**
{HISTORY}

Now generate the final PostgreSQL that could accurately solve the user issue and could be directly executed. Wrap your answer in the " ```sql
`n[Your Answer]`" ``` tag. Do not give me extra explanations or comments.

## L  BIRD-Fixer Example

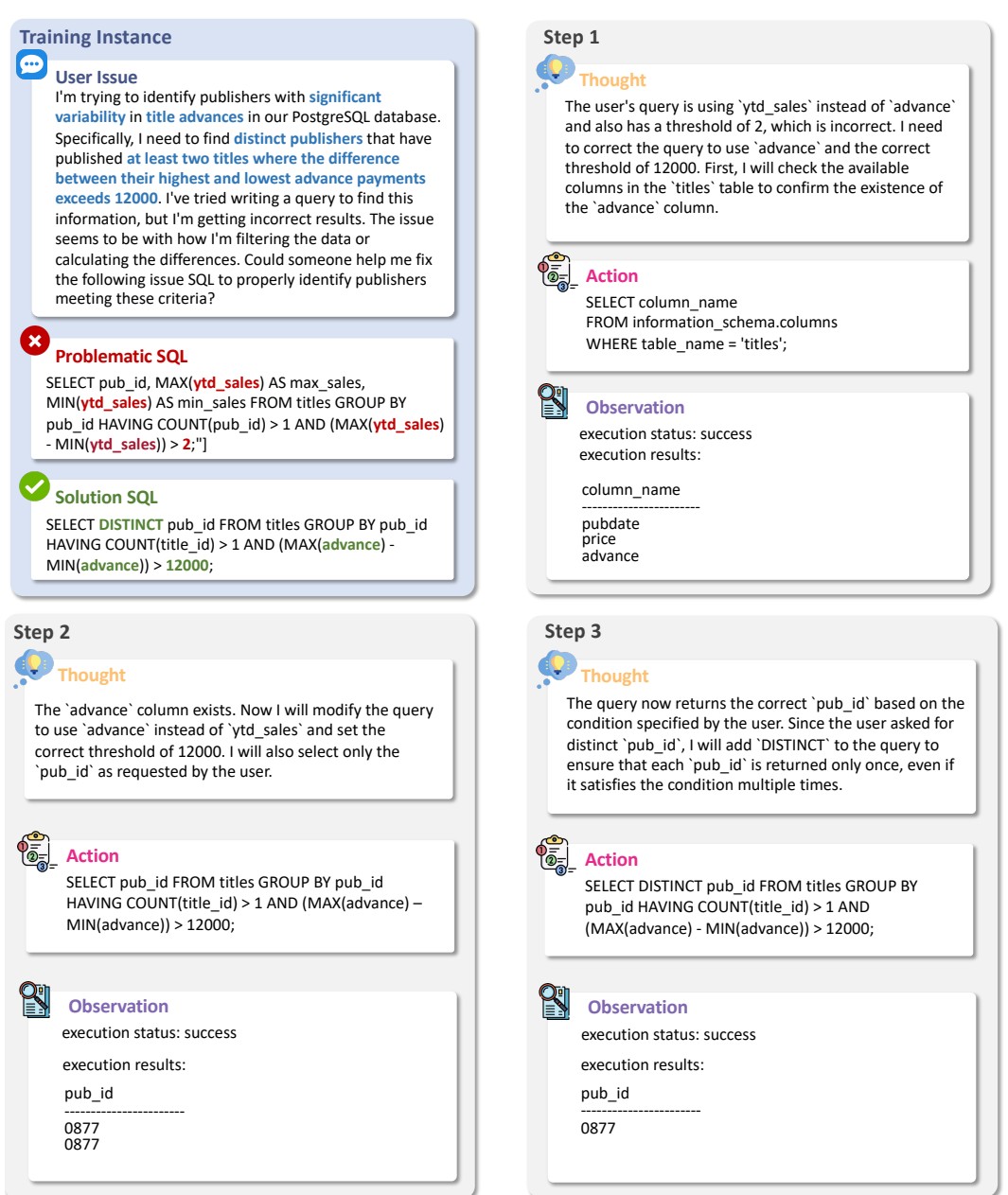

Figure 9: BIRD-Fixer Example.

# M   ƒ-Plan Example

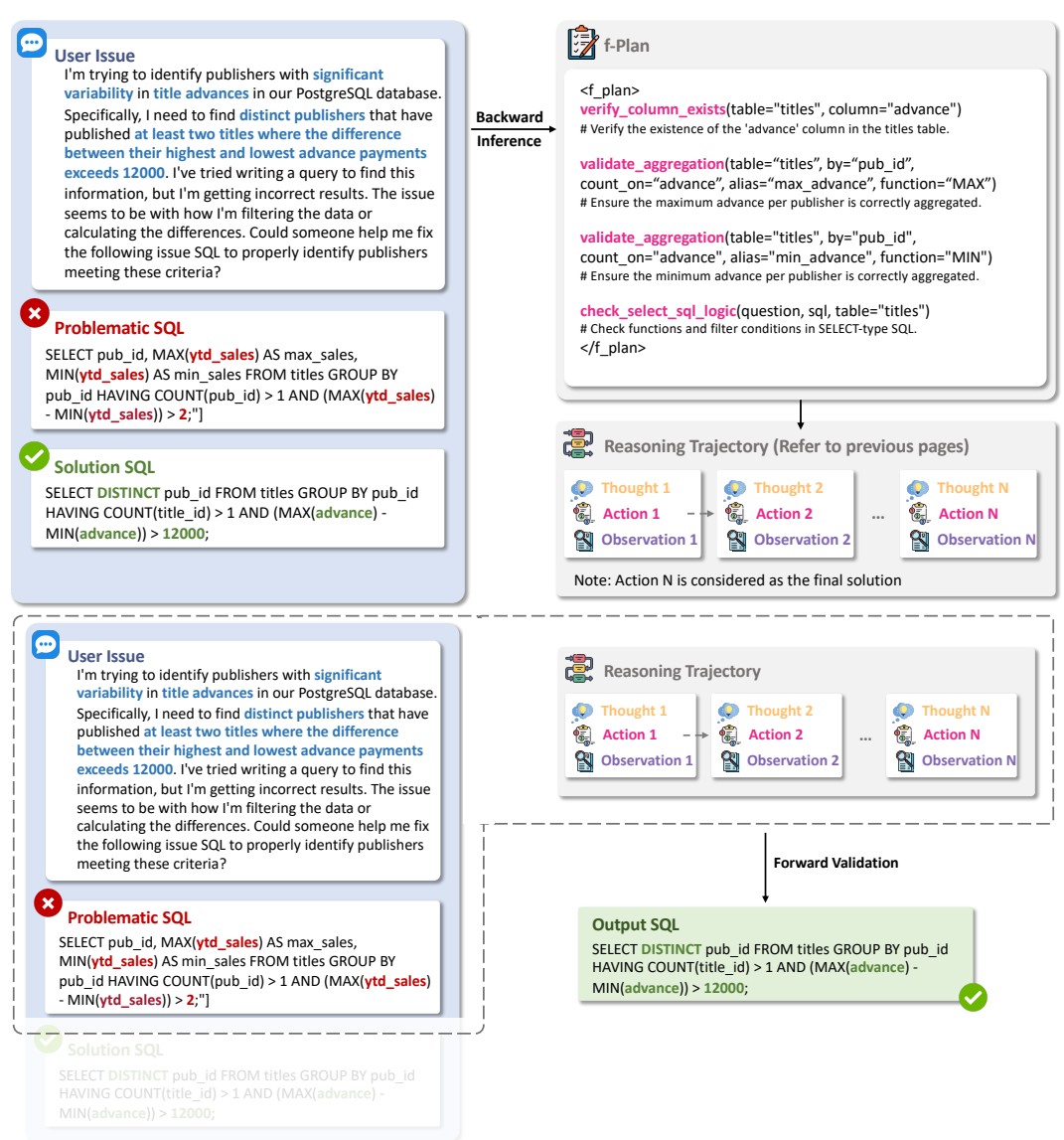

Figure 10: ƒ-Plan Example.

