# OpenReview forum: "SWE-SQL: Illuminating LLM Pathways to Solve User SQL Issues in Real-World Applications"
_NeurIPS.cc/2025/Conference — NeurIPS 2025 poster_

### Official Review · Reviewer_ixMH · 2025-06-19

**Clarity:** 3
**Significance:** 3
**Originality:** 4
**Rating:** 5
**Confidence:** 4

**Summary:**

The authors proposed a new benchmark called BIRD-CRITIC for evaluating the ability of LLMs to fix issues in SQL queries. The benchmark was constructed by sourcing real questions on SQL overflow. Human annotators then adapted and reimplemented these issues using the set of reference databases provided by the pre-existing BIRD-SQL dataset. The benchmark is challenging for recent LLMs. The authors also proposed a related method called SIX-GYM for synthetically generating fine-tuning data for this task by taking correct SQL queries, introducing errors, and generating debugging trajectories. They also developed BIRD-FIXER, an agent using an open-weights model fine-tuned using SIX-GYM, which performs on par with recent proprietary reasoning models.

**Questions:**

- Could you comment on how SQL debugging is different from Text-to-SQL generation from scratch? In particular, if you were to run an evaluation using BIRD- FIXER where the models were only given the user issue description but not the issue SQL, would the models still perform as well?
- I would expect the SQL-Act agent to struggle more on instances that require database schema migrations or destructive write operations, because the agent isn’t able to “try out” possible queries by taking actions (which would mutate the database) as it would with read-only SELECT queries. Did you observe this in your experiments?
- Did you try some of these possible alternatives for fine-tuning BIRD-FIXER?
    - Fine-tuning on a small train split of BIRD-CRITIC instead of on the SIX-GYM synthetic dataset
    - Retaining the generated functional plan in SIX-GYM instead of discarding it

**Ethical Concerns:**

["NO or VERY MINOR ethics concerns only"]

**Final Justification:**

This paper is presents a benchmark for a task that lacks existing coverage (SQL debugging) and a novel method for generating synthetic data for training models for this task. These ideas are novel, tested and practical, and overall this seems like a valuable contribution to the field.

The authors have addressed my primary concern - that SQL debugging is not significantly different from Text-to-SQL generation from scratch. The authors have ran additional experiments that show that the success rate increases significantly when the user's incorrect query is included. This demonstrates that SQL debugging is significantly different from Text-to-SQL generation from scratch, because the user's incorrect query contains important information on the user's intent. My additional questions have been sufficiently addressed as well.

**Limitations:**

Yes.

**Paper Formatting Concerns:**

None.

**Quality:**

4

**Strengths And Weaknesses:**

### Strengths

- The dataset has strong ecological validity because they are derived from real StackOverflow posts. It is also likely to be high quality due to the involvement of human annotators and experts. (Quality)
- The dataset is reimplemented to work on the reference datasets from the pre-existing BIRD-SQL dataset, which makes it easy for a user to set up the databases in their own environment for running the benchmark. (Quality)
- A number of different SQL dialects are included, and the results between them are compared. (Quality)
- Generally, the writing is clear and well-structured. (Clarity)
- The design of SIX-GYM is original and interesting. (Quality, Originality)
- SQL-fixing is a useful task with many useful applications (e.g. data scientist copilot). While a number of Text-to-SQL benchmarks already exist, there are no notable SQL-fixing benchmarks to the best of my knowledge. (Significance, Originality)

### Weaknesses

- It is not clear to me that SQL debugging is a significantly different task from Text-to-SQL generation from scratch. The analogy to code debugging is weak. While code debugging requires understanding a large amount of pre-existing code to fix a bug and then making a small change, SQL queries are typically very short, and an incorrect query can simply be rewritten from scratch. (Originality, Significance)
- Microsoft SQL Server and Oracle are proprietary databases, unlike Postgres and MySQL which are open source databases. Including SQL Server and Oracle has benefits for industry users because these are widely used databases in industry, but it may make it more difficult for academic users to run the benchmark as they may not have access to these database platforms. The paper also incorrectly claims in Section 1 Introduction that SQL Server and Oracle are open source databases. (Quality, Significance)
- I felt that the description of the SQL-Act agent in Section 5.1 is not clear enough. My understanding is that the agent’s actions are sending SQL queries, and the observations are the results of executing the queries. (Clarity)
- It would be useful to include an example of SIX-GYM in the appendix, including a function plan F and the corresponding generated solution. (Clarity)
- There are a number of minor language and typographical issues. A partial list is below. (Clarity)
    - “SQLs” should be “SQL queries” in multiple places
    - Appendix E.2 - “SQLServer” should be “SQL Server”
    - Section 5.1 SetUp - “SetUp” should be “Setup”
    - Section 6.2 Main Results - Remove the line break after “more standardized data management operations,”
    - Section 6.5 Error Analysis - The long sentence starting with “It can be concluded” should be broken into multiple sentences.
    - Section 6 Related Work and Appendix I Related Work have some repeated material.

---

> ### Author Rebuttal · Authors · 2025-07-30
>
> Thanks for your insightful and feedbacks, we will revise typo errors as you mentioned and answer your questions as below:
>
> > W1 & Q1: Could you comment on how SQL debugging is different from Text-to-SQL generation from scratch? Does BIRD-FIXER still perform more effectively without issue SQLs?
>
> **A1:** Thank you for this insightful comment. We agree that SQL queries can be short, but we think this doesn't make SQL debugging equivalent to generation from scratch or that the analogy to code debugging is weak. Our research demonstrates that SQL issue resolution is a distinct and more complex task.
>
> We summarize reasons as:
>
> 1.  **Debugging is Diagnostic, Not Just Generative:** The core difference lies in the inputs.
>     * **Text-to-SQL generation** takes a natural language description ($\mathcal{P}$) and a schema ($\mathcal{S}$) to produce a query ($\sigma_{pred}$).
>     * **Our SQL issue resolution task**, as formally defined in Section 2, takes three inputs: the description ($\mathcal{P}$), the schema ($\mathcal{S}$), and the user's own incorrect query ($\sigma_{issue}$).
>     * The presence of $\sigma_{issue}$ fundamentally changes the task. The model is not just generating a solution; it must first **diagnose the flaw** in the user's provided logic. Many issues in our BIRD-CRITIC benchmark are not syntax errors but logical or semantic flaws in which the provided SQL is often executable but produces an incorrect result. A model that simply tries to generate a query "from scratch" based on the description might easily repeat the same logical error since it hasn't understood *why* the user's initial attempt was wrong. The incorrect query provides crucial negative evidence about a plausible but flawed reasoning path. Also, we observe that most of user queries will not mention too many details about semantics but about their main aims since they think the provided SQLs could provide enough reference and only focus on error description such as which types of JOINs, or which color of cards that they construct for such recursive tree structure since issue SQL already provide these basic information.
>
> 2.  **SQL is declarative, which means Logical Complexity is High:** As we answered for Reviewer 6yZr, one of the biggest challenges in SQL debugging is due to less context, partially caused by nested and short SQL presentation. Unlike procedural languages like Python, where logic is often presented in a step-by-step manner, SQL's declarative nature nests complex reasoning within its clauses. This makes the user's intent and the query's logical flow difficult to trace from the SQL alone in a linear thinking. Furthermore, the feedback loop in database systems is often **less informative** than in general programming. For example, Oracle always return very concise error feedback such as `ORA-xxx: SQL command not properly ended`. Those in real-world applications and our benchmark often contain deeply nested logic that is not trivial to rewrite.
>     - Rewriting such a query "from scratch" is often more difficult than debugging it. The incorrect query, even with its flaws, serves as a vital blueprint of the user's intent and structural approach. Discarding it would mean losing valuable context.
>
> 3. We also conduct experiments to show performance of baselines, BIRD-FIXER for Qwen-2.5-Coder 14B and the performance of O3-mini for reference.
>
> | Model | PG w/o issue sqls | PG w/ issue sqls | MULTI w/o issue sqls | MULTI w/ issue sqls |
> |-------|-------------------------------|----------------------------------|-----------------------------------|-------------------------------------|
> | Qwen-2.5-Coder-14B | 17.74 | 31.32 | 13.33| 24.04 |
> | + BIRD-FIXER (ours) | 24.34 | 38.11 | 16.67 | 29.65 |
> | O3-mini | 26.42 | 38.87 | 15.96 | 33.33 |
>
> The sharp drop from the setting with issue SQLs to without issue SQLs demonstrate that the issue_sql is an indispensable context that allows the model to ground its reasoning and diagnose the user's specific logical error. And our method can still perform well in such setting. But of course, we could manually convert the GT SQLs into conventional text-to-SQL tasks, it may require efforts of humans or experts to make sure every detail would be contained in user queries though. We would consider this as future advanced text-to-SQL tasks. Thanks!
>
> > W2: Microsoft SQL Server and Oracle are proprietary databases, and it may make it more difficult for academic users to run the benchmark as they may not have access to these database platforms.
>
>
> **A2:** Thank you for pointing out this important clarification regarding database statement.
>
> We feel sorry for the inaccurate description in Section 1 Introduction. We will revise this to accurately describe them as community-friendly cloud-based dialect.
>
> And we want to provide more contexts here to address your concerns of Accessibility of these dialects:
>
> 1. Both SQL Server and Oracle officially provide free editions suitable for research and educational purposes. And we provide comprehensive Docker containerization for all four database environments in our submission codes. Our `docker-compose.yml` configuration automatically handles the deployment and setup, making it straightforward for both academic and industry users to reproduce our experimental environment locally, keeping special features of each dialect, such as T-SQL for SQL Server and Oracle's strict coupling between users and schemas. This eliminates the complex setup burden for researchers.
>
> 6. Researchers can still conduct meaningful evaluations using our BIRD-CRITIC-PG (PostgreSQL-only) subset while having the option to expand to multi-dialect evaluation when resources permit.
>
> In conclusion, we believe our setup provides comprehensive research value while maintaining accessibility through free developer editions and containerized deployment. Aside scientific problems that we proposed and mitigated in the paper, we also did much engineering work to make the evaluation and agent rollout faster and smooth as possible as we can by tacking many technical barriers.
>
> > W3: I felt that the description of the SQL-Act agent in Section 5.1 is not clear enough. My understanding is that the agent’s actions are sending SQL queries, and the observations are the results of executing the queries.
>
> **A3**: Yes, your understanding is correct. And as we described in the paper, BIRD-FIXER is built on SQL-ACT, so the Figure 9 in Appendix M can reveal how SQL-ACT works in an simple example. We feel sorry that we cannot include this example into main content due to limited space. But we will enhance indexing there. Thanks!
>
> > Suggestion: It would be useful to include an example of SIX-GYM in the appendix, including a function plan F and the corresponding generated solution.
>
> **A4:** Thanks, we will include these examples in Appendix if we have a chance for revision. Actually, the example of SIX-GYM is similar to real-world issue solution instances just with statistic different. We feel grateful if you can refer to first table of A1 that we answered for Reviewer Zyg2 due to space limit. We presented a comparison between tasks generated by SIX-GYM and manually annotated benchmark.
>
> > Q2: I would expect the SQL-Act agent to struggle more on instances that require database schema migrations or destructive write operations, because the agent isn’t able to “try out” possible queries by taking actions as it would with read-only SELECT queries. Did you observe this in your experiments?
>
> **A6:** Thank you for this insightful question. It's about a critical aspect of agentic design in state-altering environments.
>
> Counter-intuitively, our experiments show that the agent struggles less on instances requiring database schema migrations or destructive write operations. In fact, these "Data Management" tasks, were found to be "relatively more manageable" than complex, read-only `SELECT` queries. Our quantitative results from Figure 5 support this finding directly.
>
> The primary reason for this is the **comparative lack of diversity in the solution space**. That makes management issues while high-stakes, are "more standardized". The agent can learn these structured patterns more effectively via training techniques.
>
> However, your intuition about the agent's process is correct. We did observe that "thinking JUMP" which means agents would make abrupt leaps to final management SQL solutions without intermediate queries. This suggests the agent struggles with the inability to safely test destructive operations leading to a reasoning gap or guess. An advanced version of the agent in the future work could be equipped with a "sandbox" mode, allowing it "try out" destructive logic against reproduced databases before committing to a more accurate final solution for the actual database.
>
> > Q3:Did you try some of these possible alternatives for fine-tuning BIRD-FIXER?
>
> **A7:** Thank you for this question. We explored several alternative fine-tuning approaches:
>
> We avoided using splits of our test data for training to prevent data leakage, because BIRD-CRITIC is designed as an evaluation benchmark. So we didn't try that to make evaluation set totally unknown for training technique.
>
> We attempted with including plan generation during training smaller models to generate plans first before database interaction. However, this approach underperformed our current method for several reasons by our analysis:
>
> 1. Smaller models tend to strictly follow their initial plans even when the immediate observations suggest other directions of exploration.
> 2. Flawed initial plans lead to cascading errors throughout the debugging process.
>
> While some cases were fixed correctly through explicit planning, the overall performance decrease outweighed these successes. We believe planning remains crucial for SQL agents, but developing more robust planning mechanisms that can adapt to real-time database feedback represents an important research direction.

---

> > ### Comment · Reviewer_ixMH · 2025-08-01
> >
> > NeurIPS rebuttal
> >
> > ### A1
> >
> > Thank you for your response and additional experiments. When I wrote my original question, my hypothesis was that the initial incorrect query would have a neutral or even negative effect (i.e. serving as a distractor) on the SQL task completion rate. I appreciate your alternative explanation that the user's incorrect query contains important information on the user's intent.
> >
> > Your additional experimental results (i.e. increases of more than 10 percentage points success rate when the user's incorrect query is included) disprove my hypothesis and support your explanation. As such, I am sufficiently convinced of your explanation.
> >
> > ### A2
> >
> > Thank you for your explanation. I was not aware of the free educational edition available for SQL Server and Oracle. Given this information, I would agree with you that it is appropriate to include SQL Server and Oracle as dialects in this benchmark. I also appreciate your effort in making the benchmarking infrastructure easy to use.
> >
> > ### A3
> >
> > Noted, thank you.
> >
> > ### A4
> >
> > Thank you for this information. The table comparing the dataset summary of BIRD-CRITIC-PG and SIX-GYM is quite useful.
> >
> > ### A6
> >
> > Thank you for your additional insights. The finding that management tasks are easier is quite interesting and surprising to me.
> >
> > ### A7
> >
> > Thank you for providing information on your additional analysis. I agree that developing a better planning mechanism is an interesting future direction.

---

> ### Author Response · Authors · 2025-08-02
> **Response to Reviewer**
>
> Thank you for your thoughtful consideration of our rebuttal. We are glad to see that our responses are able to clarify the points you raised. We appreciate your constructive feedback throughout this process and are grateful for your engagement with our work!
>
> Best, \
> Authors

---

### Official Review · Reviewer_6yZr · 2025-06-28

**Clarity:** 4
**Significance:** 3
**Originality:** 3
**Rating:** 5
**Confidence:** 4

**Summary:**

This paper introduces a new and challenging SQL benchmark named BIRD-CRITIC, which covers a wide range of complex SQL tasks. It is designed to evaluate the ability of large language models (LLMs) to decompose, analyze, and reason over intricate instructions in order to generate correct SQL queries. To address common issues such as formatting or logical errors that arise from insufficient SQL understanding in current models, the authors also propose an auxiliary agent called BIRD-FIXER. This agent enhances the model’s SQL comprehension and performs post-hoc corrections on generated outputs.

**Questions:**

1. Since your work aims to address the limitations of current models on complex SQL tasks, what do you think is the main reason these models struggle? Is it due to a lack of SQL-specific knowledge, or is it because these tasks resemble mathematical reasoning and require more advanced compositional abilities? If a model were equipped with strong SQL background knowledge, would your benchmark still offer meaningful challenges and insights for further improvement?

2. Many of the tested models may not have been specifically optimized for SQL tasks. In general, SQL can be seen as one branch of code-related tasks. From your perspective, what makes SQL benchmarks necessary compared to other code benchmarks? What specific capabilities of LLMs do SQL tasks reveal that might be overlooked by more general code-oriented benchmarks?

**Ethical Concerns:**

["NO or VERY MINOR ethics concerns only"]

**Final Justification:**

I believe this is an excellent paper that not only introduces a benchmark but also offers deeper insights and innovations in the field, which contribute meaningfully to advancing the capabilities of large language models.

**Limitations:**

yes

**Quality:**

4

**Strengths And Weaknesses:**

**Strengths:**

1. The proposed benchmark, BIRD-CRITIC, includes more complex and realistic SQL tasks derived from real user queries. Current mainstream reasoning models (e.g., O3-mini) struggle with these tasks, making this benchmark a valuable resource for the community to evaluate and improve LLMs on challenging SQL problems.

2. The design of the BIRD-FIXER agent effectively corrects errors in the SQL generation process, including both syntactic and logical issues. This design makes it easier for future researchers to diagnose and improve model behavior in a more interpretable and modular way.

3. The paper is well-structured and comprehensive. It includes detailed examples of benchmark tasks, data collection methodology, dataset statistics, baseline results from popular models, ablation studies on BIRD-FIXER, and a thoughtful categorization of failure cases. The analysis is thorough and informative.

**Weaknesses:**

The approach used in BIRD-FIXER is conceptually similar to techniques already explored in other areas of LLM research, and thus lacks strong novelty. However, the adaptation of this idea to the SQL domain is still a meaningful contribution, especially given the practical challenges of SQL generation.

---

> ### Author Rebuttal · Authors · 2025-07-30
>
> > Concern 1: The approach used in BIRD-FIXER is conceptually similar to techniques already explored in other areas of LLM research, and thus lacks strong novelty. However, the adaptation of this idea to the SQL domain is still a meaningful contribution, especially given the practical challenges of SQL generation.
>
> **A1:** Thank you for raising this point. We appreciate the opportunity to clarify the novel contributions of our work.
>
> As we cited in the paper, the motivation of GYM-like dataset construction is inspired by SWE-GYM but we present an automatic framework to synthesize high-quality query-issue pairs with complex DB environment, BIRD-FIXER introduces several technical innovations:
>
> 1. Our f-Plan Boosting represents a fundamentally new approach to enhancing trajectory quality in GYM-like environments. Unlike generic fine-tuning approaches, this method leverages the logical structure inherent in SQL solutions to create richer supervisory signals. The two-phase process (backward inference → forward validation) with automatic functional plan extraction is specifically designed for SQL debugging scenarios and generated 73.7% more successful training trajectories than baseline approaches.
>
> 2. Beyond performance improvements, our method significantly enhances the efficiency of trajectory collection, as shown in our analysis:
>
> | Method | Max Tries | Successful Trajectories | BIRD-FIXER (7B) | Avg Tries | DB Execution Time (min) | Cost (USD) |
> |--------|-----------|------------------------|-----------------|-----------|-------------------------|------------|
> | Baseline | 1 | 1,254 | 25.85% | 1.0 | 306 | 8.47 |
> | f-plan (Ours) | 1 | 2,178 | 31.32% | 1.0 | 324 | 27.44 |
> | Reject Sampling | 5 | 1,910 | 29.06% | 4.2 | 1377 | 108.05 |
> | Reject Sampling + f-plan | 5 | 2,560 | 33.02% | 1.7 | 810 | 41.16 |
>
> **Trajectory Sampling Strategy:**
> - **Baseline**: Standard SQL-ACT rollout with single attempt
> - **f-plan (Ours)**: Our f-plan approach (single attempt)
> - **Reject Sampling**: Multiple attempts (max_tries=5) with SQL-ACT by setting temperature=0.8, and early stop if the trajectory is successful.
> - **Reject Sampling + f-plan**: Combined approach of f-plan boosting with rejection sampling.
>
> Our approach achieves more successful trajectories generated by dramatically lower computational cost compared to standard rejection sampling methods. Also, it shows f-Planning can arrive at a successful trajectory much sooner than regular and popular trajectory augmentation strategy as rejection sampling as Reviewer Zyg2 mentioned. To our best knowledge, this is the first work to propose such a strategy for enhancing utilization of GYM-like datasets. We think this would open important directions for better utilization of GYM-like datasets and hope it will inspire further research in efficient trajectory generation methods for specialized domains.
>
> 3. Our SQL-ACT framework treats arbitrary SQL commands as actions rather than restricting to predefined tools, fundamentally expanding the action space beyond conventional tool-based agents. Experimental results demonstrate that SQL-driven agents can handle more complex debugging scenarios through this flexible exploration space, which is essential for addressing the diverse and unpredictable nature of user errors and varied database logic requirements.
> 4. We also present a An innovative decoupling mechanism, **Generative Thought Mode (GTM)**, that separates debugging reasoning from SQL code generation, addressing overfitting issues while maintaining the model's native understanding of diverse SQL dialects.
>
> Thanks for your question and suggestion.
>
>
> > Question 1: what do you think is the main reason these models struggle? Is it due to a lack of SQL-specific knowledge, or is it because these tasks resemble mathematical reasoning and require more advanced compositional abilities?
>
> **A2:** We sincerely thank you for proposing such deep questions. Actually, we agree that the primary difficulty for Large Language Models (LLMs) in complex SQL tasks stems more from limitations in their reasoning and compositional abilities than from a mere lack of SQL-specific knowledge.
>
> 1. First, as we can observe, these models perform well on popular text-to-SQL benchmarks such as BIRD-SQL, Spider, etc. For example, Qwen-2.5-Coder can achieve 58.4% EX (which is a more strict metric) on BIRD without any fine-tuning and prompting techniques, but 30.75% success rate (by a more flexible test case evaluation) in BIRD-CRITIC-PG and 24.74% in BIRD-CRITIC-MULTI, which shows even though they have a good knowledge about SQLs, actually, SQL debugging logics maybe still a new research problem.
> 2. The process of debugging SQL is inherently a "reasoning-driven" task. It requires not only understanding the user's intent from a verbose description but also analyzing the underlying query logic, identifying subtle errors, and interacting intensively with the database schema in Line 46-49. For example, an analysis of the errors made by these models reveals that "Incorrect Logic" is the most frequent failure category. This points to fundamental misunderstandings of data structures or how to perform complex transformations. Therefore, merely equipping a model with more SQL background knowledge would not be sufficient but a pre-requisite. SQL debugging requires a multi-dimension combination of capabilities: domain-specific knowledge (including mathematical computation and specialized fields like chemistry when working with scientific databases), deep understanding of data relationships and schema semantics, and sophisticated agentic debugging reasoning similar to what we observe in SWE-Bench. We will elaborate on these interconnected requirements in the following detailed response.
>
> > Question 2: In general, SQL can be seen as one branch of code-related tasks. From your perspective, what makes SQL benchmarks necessary compared to other code benchmarks? What specific capabilities of LLMs do SQL tasks reveal that might be overlooked by more general code-oriented benchmarks?
>
> **A2:** Thanks for this deep question. We think the special challenges by SQL debugging would be:
>
>
> 1. A key challenge that SQL benchmarks illuminate is the capability of debugging **declarative** code. Unlike procedural languages like Python, where logic is often presented in a step-by-step manner, SQL's declarative nature nests complex reasoning within its clauses. This makes the user's intent and the query's logical flow difficult to trace from the code alone in a linear thinking. Furthermore, the feedback loop in database systems is often **less informative** than in general programming. For example, Oracle always return very concise error feedback such as `ORA-xxx: SQL command not properly ended`. This error confirms a syntax problem but gives no clue as to *where* or *why*, forcing the model to deduce the issue is a non-standard keyword making it a hard way of error location. This lack of explicit guidance means the model cannot simply follow an error stack trace; it must adopt an **agentic, exploratory approach** to form and test hypotheses against the database. While CTEs in SQL can break down queries step-by-step, they sacrifice precise Query Execution Plan generation, preventing accurate efficiency analysis. As we know that the SQL efficiency is important concern in DB field. Therefore, many user issue SQLs stem from declarative constructs that inherently hide useful reasoning and diagnostic clues.
>
> 2. As we showed in error analysis, Chain-of-Errors matter in SQL debugging which requires agent capable of deep, causal reasoning in the face of masked errors. To be specific, the database engine reports the most superficial issue, masking a deeper, dependent error that is the true root cause. For instance, a type mismatch error might be reported, but the underlying problem could be an incorrect join that brought together the wrong columns in the first place. This contrasts with many general code benchmarks where an error trace can provide a clearer path to the initial point of failure.
>
> 3. As usual Text-to SQL tasks, SQL queries are inherently business-driven, requiring extensive domain-specific knowledge across various fields ( in our benchmark it can be from financial calculations to chemical compound relationships). Models must not only possess this external knowledge but also map it accurately to specific database representations involving **precise** tables, columns, and data types. This semantic bridging between domain concepts and database schema is a unique challenge rarely found in general programming tasks. The first example in Figure 8 demonstrates how domain understanding directly impacts query correctness.
>
> In conclusion we can consider SQL debugging a specialized complex tasks that are dictated by particularly their requirements for unique nature of databases, particularly their requirements for privacy, efficiency, data integrity, and domain knowledge mapping. An LLM cannot succeed by simply converting text; it must operate as a capable of exploration, hypothesis, inference, and adaptation within this constrained, stateful environment.

---

> ### Comment · Reviewer_6yZr · 2025-08-07
>
> Thank you for addressing my question. I believe this is an excellent paper that not only introduces a benchmark but also offers deeper insights and innovations in the field, which contribute meaningfully to advancing the capabilities of large language models. I will maintain my positive score.

---

> ### Author Response · Authors · 2025-08-07
> **Thank you for your Reviews and Time**
>
> Thanks for your appreciation of our work! And it encouraging to see that our further responses can answer your questions. Thank you for sharing your feedback and insights with us!

---

### Official Review · Reviewer_Zyg2 · 2025-07-02

**Clarity:** 4
**Significance:** 3
**Originality:** 3
**Rating:** 5
**Confidence:** 4

**Summary:**

This paper presents first a benchmark (BIRD-CRITIC) which deals with repairing and debugging SQL-based programming problems. The authors manually curated problems from real user issues after applying them on a controlled environment. The authors also created a synthetic training dataset by synthetically generating new problems and solution using an LLM. The paper performs several rounds of evaluation and demonstrate that prior popular LLMs still struggle on the task of SQL-based debugging and shows the effect of fine-tuning on the synethtic training dataset has on performance

**Questions:**

1. Did the authors manually evaluate the quality of the trajectories/problems in the synthetic benchmark and see if they are of similar quality to the collected benchmark?
2. What exactly are the benefits of f-Plan Boosting (i.e., Can it be used to generated very difficult trajectories from problems that the basic approach cannot?). Furthermore, what is the trajectory generation setting and does it use some level of rejection sampling scheme?

**Ethical Concerns:**

["NO or VERY MINOR ethics concerns only"]

**Final Justification:**

In my opinion I think the paper does extensive work not only to create an environment that supports training SQL agents (generating synthetic training data) but additionally purpose a simple method to improve the probability and success rate of training trajectries. Furthermore the basic agent setup proposed in the paper can also serve as a good baseline for future work to compare with. Additionally, the authors also answered my concerns and questions with additioanly experiments and text that I hope they will include in the final version. As such I remain positive on this work and would recommend acceptance.

**Limitations:**

Yes

**Paper Formatting Concerns:**

No concerns

**Quality:**

3

**Strengths And Weaknesses:**

Overall the paper is well-written and the amount of contribution is strong

strengths
- extensive description of how the benchmark is constructed
- human-validated dataset construction with cross-validation to ensure high accuracy of the benchmark
- the high number of testcases per each problem is also highly encouraging
- proposed additional synthetic training dataset that can be further adopted and used by future work
- strong amount of evaluation to analyze across different problem types and across many different models and model/agent setups

weaknesses
- its bit unclear what the exact quality of the synthetic dataset generated to create SIX-GYM is. For example, it would benefit the work if the authors selected several samples from the synthetic dataset and manually verified the quality of the benchmark
- "In our experiments, running Gemini-2.0-Flash with SQL-ACT on SIX-GYM produces just 1,254 successful trajectories" from this sentence it is very unclear if the authors performed some level of rejection-sampling or just simple sample the model once?
- The previous point also follows my next question: what exactly are the benefits of f-Plan Boosting? For example the authors claims that without f-Planning not all the trajectories can be successfully generated in the synthetic benchmark. However, is this due to the low number of sampled trajectories (i.e., lack of rejection sampling). I think a better benefit would be to show that f-Planning can arrive at a successful trajectory much sooner than regularly generating a trajectory. However this analysis is not shown in the paper

---

> ### Author Rebuttal · Authors · 2025-07-30
>
> Thanks for your appreciation of our work and thoughtful suggestions and feedbacks. Here are our responses:
>
> > Concern 1: It's bit unclear what the exact quality of the synthetic dataset generated to create SIX-GYM is. For example, it would benefit the work if the authors selected several samples from the synthetic dataset and manually verified the quality of the benchmark.
>
> **A1:** Thank you for raising this. We acknowledge that demonstrating the quality of SIX-GYM is crucial for establishing the credibility of our approach.
>
> Originally, The downstream performance improvements achieved by models trained on SIX-GYM serve as a strong indicator of dataset quality. BIRD-FIXER demonstrates consistent gains across different model architectures, suggesting that our synthetic data captures meaningful debugging patterns that transfer effectively to real-world scenarios.
>
> Beyond that, we also present detailed statistics comparing SIX-GYM with manually curated BIRD-CRITIC to demonstrate comparable data characteristics below. It shows that our synthesized GYM has similar quality and complexity with human annotated data:
>
> | Dimension | BIRD-CRITIC-PG (Benchmark) | SIX-GYM (SQL-Rewind) |
> |-----------|-------------|---------|
> | User Query Length (mean/max) | 162.98/1046 | 171.1/882 |
> | Issue SQL Length (mean/max) | 133.29/1262  | 110.2/1089 |
> | Solution SQL Length (mean/max) |112.64/853 | 94.8/772 |
> | SQL Keywords Coverage | 165 | 157 | Better feature coverage |
> | Complex Operations (JOINs, Window Funcs) % | 54.5 | 54.3 |
> | Multi-clause Queries % | 59.4 | 61.2 |
> | SQL Diversity Ratio | 0.728 | 0.750 |
> * Diversity Ratio = Unique 3-grams / Total 3-grams of SQLs; The closer the ratio is to 1.0, the higher the diversity.
>
> Also, we sampled and dispatched 100 medium-difficult tasks to 5 annotators to manually evaluate synthesized data quality as you suggested:
> - **Issue-Solution Logical Consistency (88%)**: This measures whether the synthetic issue SQL can logically lead to the provided solution SQL. 0 means the transformation is arbitrary or nonsensical or it contains multi-hop gaps (not only one type of errors), while 1 means the debugging path is logically sound and semantically coherent.
> - **Realistic Error Patterns (94%)**: This evaluates whether the introduced errors mirror authentic debugging scenarios commonly encountered in their experience of understanding real-world user issues during testing annotation. 0 means the errors are obviously artificial or not meaningful, while 1 means they reflect genuine mistakes developers make in practice.
> - **Executable Code Quality (100%)**: This verifies that the solution SQL executes successfully against the target database schemas. 0 means the solution SQL contains syntax errors or runtime failures, while 1 means both queries execute without errors.
> - **User Intent Coherence (86%)**: This measures whether the generated user issue description accurately reflects the debugging objective, provides sufficient context, and has less ambiguity. 0 means the description is unclear or misaligned with the SQL changes, while 1 means it clearly conveys the intended debugging goal.
>
> Thus, we think the combination of strong empirical results, comparable statistical distributions to manual benchmarks, demonstrates that SIX-GYM achieves high-quality synthetic data generation suitable for training effective SQL debugging agents. Thanks for your question!
>
>
>
>
>
> > Concern 2: What exactly are the benefits of f-Plan Boosting (i.e., Can it be used to generated very difficult trajectories from problems that the basic approach cannot?). Furthermore, what is the trajectory generation setting and does it use some level of rejection sampling scheme?
>
> **A2:** Thank you for this excellent question. Your inquiry touches on a critical aspect of our contribution that deserves detailed clarification.
>
> Database agent training paradigm presents unique challenges due to its heterogeneous and long-context nature. Each trajectory requires extensive interaction with database environments, making data collection particularly time-consuming, especially when working with large databases
>
> To better answer your question, we conducted additional experiments comparing different trajectory generation strategies:
>
> | Method | Max Tries | Successful Trajectories | BIRD-FIXER (7B) | Avg Tries | DB Execution Time (min) | Cost (USD) |
> |--------|-----------|------------------------|-----------------|-----------|-------------------------|------------|
> | Baseline | 1 | 1,254 | 25.85% | 1.0 | 306 | 8.47 |
> | f-plan (Ours) | 1 | 2,178 | 31.32% | 1.0 | 324 | 27.44 |
> | Reject Sampling | 5 | 1,910 | 29.06% | 4.2 | 1377 | 108.05 |
> | Reject Sampling + f-plan | 5 | 2,560 | 33.02% | 1.7 | 810 | 41.16 |
>
> **Trajectory Sampling Strategy:**
> - **Baseline**: Standard SQL-ACT rollout with single attempt
> - **f-plan (Ours)**: Our f-plan approach (single attempt)
> - **Reject Sampling**: Multiple attempts (max_tries=5) with SQL-ACT by setting temperature=0.8, and early stop if the trajectory is successful.
> - **Reject Sampling + f-plan**: Combined approach of f-plan boosting with rejection sampling (max_tries=5).
>
> **Complex Issue SQL Resolution Analysis:**
>
> We analyzed instances where f-plan succeeds but baseline and rejection sampling fail, finding that f-plan particularly excels on complex SQL structures:
>
> | Issue SQL Complexity | Baseline Success Rate | Reject Sampling Success Rate | f-plan Success Rate |
> |----------------|----------------------|----------------------------|-------------------|
> | Simple (1-2 clauses) | 52.3% | 62.8% | 70.3% |
> | Medium (3-4 clauses) | 38.7% | 58.8% | 69.4% |
> | Complex (5+ clauses) | 19.4% | 37.1% | 66.3% |
> | High Keyword Diversity (10+ unique) | 24.1% | 35.6% | 54.3% |
> | Nested Operations (2+ levels) | 21.8% |  36.8% | 49.2% |
>
> In summary, we can conclude the benefits of f-plan:
>
> 1. It's a **cost-efficient** trajectory-augmentation strategy leading to better agent performance in complex DB environments with less time and token/money cost.
>
> 2. **Execution Efficiency**: Reduces average tries from 4.2 to 1.7 when combined with rejection sampling, significantly reducing database execution time and computational overhead. And this prove your guess that our f-plan boosting can make rejection sampling reach success sooner, we would include this into paper if we get a chance, thank you!
>
> 3. **Complex Problem Resolution**: We observe that f-plan can resolve GYM instances that both rejection sampling and baseline cannot handle. F-plan helps teacher LLMs resolve instances with complex issue SQLs, showing obvious improvement when SQL contains more clauses and diversity of keywords (representing more diverse user intents).
>
> 4. **Plug-and-Play Compatibility**: F-plan is orthogonal to traditional trajectory-augmentation strategies like rejection sampling, providing additional 34.0% successful trajectory improvement when combined while maintaining efficiency gains.
>
> Thanks for your insightful question. After answering with additional experiments as evidence, it further demonstrates the benefits of our method for complex database debugging scenarios. We would include this analysis into paper to make readers better understand its benefits beyond downstream performance

---

> > ### Comment · Reviewer_Zyg2 · 2025-08-04
> >
> > Thanks to the authors for providing a detail reponse to my concerns and questions. In particular thanks for adding the manual analysis into the synethtic data generated. I hope the authors may add them to the appendix of the final paper. Furthermore I also understand better the contribution and benefit of f-planning with the new experiments. I highly encourage the authors to add the experiment and description into the work as it helped me clarify the outcome of the approach. As such, I remain positive on this paper.

---

> ### Author Response · Authors · 2025-08-04
>
> Thank you for your appreciation of our work and for posting insightful questions and suggestions. We will incorporate these into our revision if given the chance. And we are pleased that our rebuttal provided helpful clarification to you. Thanks!

---

### Official Review · Reviewer_HbCB · 2025-07-03

**Clarity:** 2
**Significance:** 3
**Originality:** 3
**Rating:** 5
**Confidence:** 5

**Summary:**

The contributions of this paper are fourfold: (1) BIRD-CRITIC Benchmark: A comprehensive benchmark for SQL issue debugging, which includes real-world tasks and custom evaluation scripts. (2) SIX-GYM Environment: An automated training environment that generates large-scale, executable issue-solution datasets by employing the SQL-Rewind strategy. (3) f-Plan Boosting: A method that extracts high-level debugging plans from SQL solutions, thereby significantly increasing the number of successful training trajectories. (4) BIRD-FIXER Agent: An open-source agent that leverages SQL-ACT and f-Plan Boosting, achieving state-of-the-art performance on SQL debugging tasks.
By providing a method for constructing training data, this paper reduces the consumption of human and material resources in data construction. Additionally, the proposed benchmark allows for the evaluation of LLMs' capabilities in SQL issue debugging, thereby contributing to the advancement of debugging capabilities in this area.

**Questions:**

1. Whether the training data for effective reasoning model?
2. 2. BIRD-FIXER was only fine-tuned on PostgreSQL trajectories. If fine-tuned on multi-language trajectories, could the model capabilities be further enhanced?

**Ethical Concerns:**

["NO or VERY MINOR ethics concerns only"]

**Final Justification:**

Taking all the discussions into consideration, I agree to accept this paper.

**Limitations:**

yes

**Quality:**

3

**Strengths And Weaknesses:**

Strengths:
1. Current research in the Text-to-SQL field mainly focuses on evaluating the ability of large language models (LLMs) to accurately convert natural language queries into SQL statements. However, there is still a lack of corresponding datasets for the evaluation of LLMs' SQL debugging capabilities. The dataset proposed in this paper fills the gap in this field in this aspect.
2. This dataset is reasonably constructed and based on the relational database in the BIRD-SQL development set. This database has a complex structure and can simulate the real-world database environment to a certain extent. The annotation process was accomplished through the collaboration of two professional teams, and the annotation quality was ensured through cross-validation. The manual annotation method was adopted to further enhance the accuracy and reliability of the data.
3. The set task is highly challenging. Despite the continuous improvement of LLMs capabilities, the BIRD-CRITIC dataset still poses significant challenges to existing models. For instance, the best-performing model O3-Mini-2025-01-31 has a success rate of only 38.87% in the PostgreSQL subtask and 33.33% in the multi-dialect subtask, indicating considerable room for improvement.
4. Furthermore, the article also proposes a new method for automatically constructing training data annotations, and through experiments, it is verified that this method has a significant effect in improving the SQL debugging ability of LLMs.

Weakness：
1. The experimental section could be more robust. Although this paper claims that the proposed training method is model-architecture-agnostic, it only trains four models, and the training does not cover the thinking model. This is not sufficient to demonstrate the effectiveness of the training data.
2. The paper mentions the differential distributions of SQL dialects within the respective training corpora of these models, which has an impact on the models' performance across different dialects. However, in the paragraph at line 300, it is stated that the model is only trained on PostgreSQL and is extensible, but there are no experiments to show whether training on other dialects would lead to improvements.

---

> ### Author Rebuttal · Authors · 2025-07-30
>
> We sincerely thanks for your appreciation of our work. And are grateful for your insightful questions. Let us answer them one by one:
>
> > Concern 1: Although this paper claims that the proposed training method is model-architecture-agnostic, it only trains four models, and the training does not cover the thinking model. Whether the training data for effective reasoning model?
>
> **A1:** We appreciate your attention to the breadth of our experimental details!
> Regarding model architecture diversity, our selection was strategically designed to represent key and popular architectural paradigms in the current LLM landscape. We evaluated models across different parameter scales (7B-14B), training methodologies (general pre-training vs. code-specialized), and architectural families (Llama, Qwen, Phi) with their different distributions of pre-trained data and model structures. The consistent improvements observed across these diverse architectures provide evidence for the architecture-agnostic nature of our approach.
>
> Regarding reasoning model, actually, the open-weights appear very recently, i.e., Qwen 3 with reasoning choice is in 14th May, and Phi-4-Reasoning appears on 30th April, which are closer to submission DDL. However, we appreciate your suggestions for this, and we tried training with the same trajectory data for Qwen-3, with reasoning enabled during inference. The result shows:
> | Category | Query  | Management  | Personalization | Overall |
> |----------|-------------|-----------------|----------------------|---------|
> | Qwen3-8B-Reason | 27.49 | 39.77 | 21.85 | 27.92 |
> | +BIRD-FIXER | **28.87** | **52.27** | **26.49** | **32.08** |
>
> It shows our method can be also effective to recent reasoning-based open-weights model, and we thanks for your suggestion to make our statement more rigorous and robust.
>
> > Question: BIRD-FIXER was only fine-tuned on PostgreSQL trajectories. If fine-tuned on multi-language trajectories, could the model capabilities be further enhanced?
>
> **A2:** Thank you for raising this important point. Database agent training paradigm presents unique challenges due to its heterogeneous and long-context nature. Each trajectory requires extensive interaction with database environments, making data collection particularly time-consuming, especially when working with large databases. Additionally, while executable SQLs after processing are available on platforms like StackOverflow, reconstructing their corresponding database environments remains technically challenging. These constraints motivated our design of methods that account for such low-resource scenarios while demonstrating generalized capabilities. However, we agree with your suggestion which can further strengthen our statement. Within limited time and budget during rebuttal, we conducted preliminary experiments by collecting 2,216, 1,432, and 1,986 multi-dialect SIX-GYM instances for MySQL, Oracle, and SQL Server respectively, following the methods described in Sections 4 and 5 to fine-tune Qwen-2.5-Coder-7B specifically for each dialect:
>
> | Model | MySQL | Oracle | SQL Server |
> |-------|-------|--------|------------|
> | Base | 18.37 | 7.14 | 20.41 |
> | + BIRD-FIXER | **27.55** | **12.24** | **29.59** |
>
> These results demonstrate obvious improvements with dialect-specific training. Notably, we observe that our previous generalization approach primarily addressed syntax and logic errors leveraging shared database knowledge across dialects. However, dialect-specific training enables the solution of environment-specific errors. For example, for SQL Server specifically, this includes handling T-SQL-specific constructs such as error handling with TRY...CATCH blocks. These dialect-specific nuances require targeted training to master more effectively.
>
> This additional investigation validates your hypothesis and represents a promising direction for future work. We appreciate your insightful question.

---

> > ### Comment · Reviewer_HbCB · 2025-08-07
> >
> > The authors have addressed my concerns, but I had originally recommended acceptance of the paper, so my rating will remain unchanged.

---

> ### Author Response · Authors · 2025-08-07
> **Thanks for your Comments!**
>
> We are grateful for your acknowledgment of our work and pleased that our responses can address your concerns. We value your feedback and the time you invested in this process! Thanks!

---

### Decision · Program_Chairs · 2025-09-17

**Decision:**

Accept (poster)

**Comment:**

The contributions are: (1)  A comprehensive benchmark for SQL issue debugging, which includes real-world tasks and custom evaluation scripts,  (2)  an automated training environment that generates large-scale, executable issue-solution datasets, (3) a method that extracts high-level debugging plans from SQL solutions and increases the number of successful training trajectories, and  (4) an open-source agent that achieves state-of-the-art performance on SQL debugging tasks.

The authors addressed all concerns of the reviewers in their rebuttal and all reviewers unanimously voted for acceptance. The additional arguments and experiments presented in the rebuttal should be included in the revised version of the manuscript.